# Experimental Designs for Heteroskedastic Variance

**Justin Weltz**[*] **Tanner Fiez**[†] **Eric Laber**[‡] **Alexander Volfovsky**[§]
**Blake Mason** [¶] **Houssam Nassif** [‖] **Lalit Jain** [**]

## Abstract

Most linear experimental design problems assume homogeneous variance, even though heteroskedastic noise is present in many realistic settings. Let a learner have access to a finite set of measurement vectors $\mathcal{X} \subset \mathbb{R}^d$ that can be probed to receive noisy linear responses of the form $y = x^\top \theta^* + \eta$. Here $\theta^* \in \mathbb{R}^d$ is an unknown parameter vector, and $\eta$ is independent mean-zero $\sigma_x^2$-strictly-sub-Gaussian noise defined by a flexible heteroskedastic variance model, $\sigma_x^2 = x^\top \Sigma^* x$. Assuming that $\Sigma^* \in \mathbb{R}^{d \times d}$ is an unknown matrix, we propose, analyze and empirically evaluate a novel design for uniformly bounding estimation error of the variance parameters, $\sigma_x^2$. We demonstrate the benefits of this method with two adaptive experimental design problems under heteroskedastic noise, fixed confidence transductive best-arm identification, and level-set identification; proving the first instance-dependent lower bounds in these settings. Lastly, we construct near-optimal algorithms and empirically demonstrate the large improvements in sample complexity gained from accounting for heteroskedastic variance in these designs.

## 1 Introduction

Accurate data analysis pipelines require decomposing observed data into signal and noise components. To facilitate this, the noise is commonly assumed to be additive and homogeneous. However, in applied fields this assumption is often untenable, necessitating the development of novel design and analysis tools. Specific examples of deviations from homogeneity include the variability in sales figures of different products in e-commerce [24, 31], the volatility of stocks across sectors [5], spatial data [58, 43], and genomics [48, 10]. This heteroskedasticity is frequently structured as a function of observable features. In this work, we develop two near-optimal adaptive experimental designs that take advantage of such structure.

There are many approaches, primarily developed in the statistics and econometrics literature, for efficient mean estimation in the presence of heteroskedastic noise (see [56, 16] for a survey). These methods concentrate on settings where data have already been collected and do not consider sampling to better understand the underlying heteroskedasticity. Some early work exists on experimental design in the presence of heteroskedastic noise, however these methods are non-adaptive (see Section 2 for details). Importantly, *using naive data collection procedures designed for homoskedastic environments may lead to sub-optimal inference downstream* (see Section 1.2 for examples). A scientist may fail to take samples that allow them to observe the effect of interest with a fixed sampling budget, thus wasting time and money. This work demonstrates how to efficiently and adaptively collect data

---

[*]Department of Statistical Science, Duke University, justin.weltz@duke.edu, work conducted at Amazon

[†]Amazon.com, USA, fieztann@amazon.com

[‡]Department of Statistical Science, Duke University, eric.laber@duke.edu

[§]Department of Statistical Science, Duke University, alexander.volfovsky@duke.edu

[¶]Amazon.com, USA, bjmason@amazon.com

[‖]Meta, USA, houssamn@meta.com

[**]Michael G. Foster School of Business, University of Washington, lalitj@uw.edu, work conducted at Amazon

37th Conference on Neural Information Processing Systems (NeurIPS 2023).

in the presence of heteroskedastic noise. It bridges the gap between heteroskedasticity-robust (but potentially inefficient) passive data collection techniques, and powerful (but at times brittle) active algorithms that largely ignore the presence of differing variances.

## 1.1 Problem Setting

To ground our discussion of heteroskedasticity in active data collection, we focus on adaptive experimentation with covariates; also known as pure-exploration linear bandits [7, 45]. Let a learner be given a finite set of measurement vectors $\mathcal{X} \subset \mathbb{R}^d$, $\mathrm{span}(\mathcal{X}) = \mathbb{R}^d$, which can be probed to receive linear responses with additive Gaussian noise of the form:

$$y = x^\top \theta^* + \eta \quad \text{with} \quad \eta \sim \mathcal{N}(0, \sigma_x^2), \quad \sigma_x^2 := x^\top \Sigma^* x. \tag{1}$$

Here, $\theta^* \in \mathbb{R}^d$ is an unknown parameter vector, $\eta$ is independent mean-zero Gaussian noise with variance $\sigma_x^2 = x^\top \Sigma^* x$, and $\Sigma^* \in \mathbb{R}^{d \times d}$ is an unknown positive semi-definite matrix with the assumption that $\sigma_{\min}^2 \leq \sigma_x^2 \leq \sigma_{\max}^2$ for all $x \in \mathcal{X}$, where $\sigma_{\max}^2, \sigma_{\min}^2 > 0$ are known. We focus on Gaussian error for ease of exposition but extend to strictly sub-Gaussian noise in the Appendix. The linear model in Eq. (1) with unknown noise parameter $\Sigma^*$ directly generalizes the standard multi-armed bandit to the case where the arms have different variances [4]. Additionally, this heteroskedastic variance structure arises naturally in linear mixed effects models [20], which are used in a variety of experimental settings to account for multi-level heterogeneity (patient, spatial and block design variation [41]). Throughout, $\mathcal{Z} \subset \mathbb{R}^d$ will be a set (that need not be equal to $\mathcal{X}$) over which our adaptive designs will be evaluated. While the theory holds true for $\sigma_{\min}^2 \leq \sigma_x^2 \leq \sigma_{\max}^2$ in general, we consider the case where $\sigma_{\max}^2 = \max_{x \in \mathcal{X}} \sigma_x^2$ and $\sigma_{\min}^2 = \min_{x \in \mathcal{X}} \sigma_x^2$ so that $\kappa := \sigma_{\max}^2 / \sigma_{\min}^2$ summarizes the level of heteroskedasticity.

## 1.2 Heteroskedasticity Changes Optimal Adaptive Experimentation

To underscore the importance of adapting to heteroskedasticity, we consider the performance of RAGE [12], a best arm identification algorithm designed for homoskedastic settings. This algorithm does not account for differing variances, so it is necessary to upper bound $\sigma_x^2 \leq \sigma_{\max}^2$ for each $x$ to identify the best arm with probability $1 - \delta$ for $\delta \in (0, 1)$. In Fig. 1, we compare this approach to the optimal sampling allocation accounting for heteroskedasticity in a setting that is a twist on a standard benchmark [46] for best-arm identification algorithms. Let $\mathcal{X} = \mathcal{Z} \subset \mathbb{R}^2$ and $|\mathcal{X}| = 3$. We take $x_1 = e_1$ and $x_2 = e_2$ to be the first and second standard bases and set $x_3 = \{\cos(0.5), \sin(0.5)\}$. Furthermore, we take $\theta^* = e_1$ such that $x_1$ has the highest reward and $x_3$ has the second highest. In the homoskedastic case, an optimal algorithm will focus on sampling $x_2$ because it is highly informative for distinguishing between $x_1$ and $x_3$ as shown in Fig 1b. However, if the errors are heteroskedastic, such a sampling strategy may not be optimal. If $\sigma_2 \gg \sigma_1, \sigma_3$, the optimal allocation for *heteroskedastic* noise focuses nearly all samples on $x_1$ and $x_3$ (as shown in Fig. 1a), which are more informative than $x_2$ because they have less noise. Hence, ignoring heteroskedasticity and upper-bounding the variances leads to a suboptimal sampling allocation and inefficient performance. This effect worsens as the amount of heteroskedasticity increases, leading to an additional multiplicative factor of $\kappa$ (cf., [12], Thm. 1) as demonstrated in Fig. 1c, In this important example, we see that the sample complexity of an algorithm that ignores heteroskedasticity suffers a linear dependence on $\kappa$, while a method that adapts to the heteroskedasticity in the data does not.

## 1.3 Paper Contributions

1. **Finite Sample Noise Estimation Guarantees.** We use an experimental design technique to construct estimates of the heteroskedastic noise variances with bounded maximum absolute error. This adaptive experimental design is a two-phase sample splitting procedure based on a pair of distinct G-optimal experimental designs [28], with bounds that scale favorably with the underlying problem dimension.

2. **Near-Optimal Experimental Design Algorithms.** We propose adaptive experimental design algorithms for transductive linear best-arm and level-set identification with heteroskedastic noise, that combine our variance estimation method with optimal experimental design techniques. We prove sample complexity upper bounds for unknown $\Sigma^*$ that nearly match instance dependent lower bounds when $\Sigma^*$ is known. Importantly, we show that in contrast to naive methods which

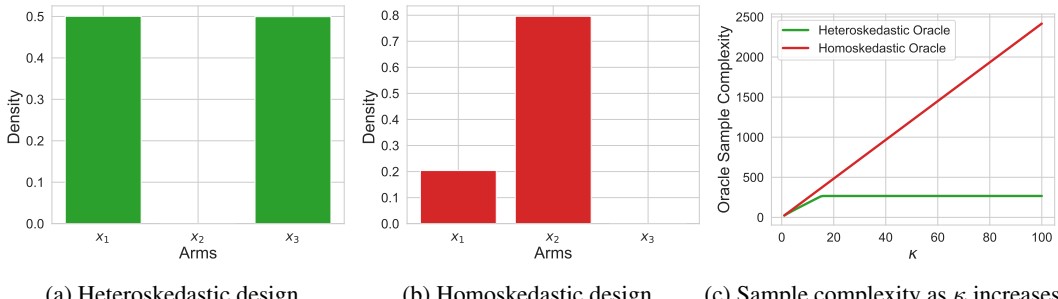

| (a) Heteroskedastic design | (b) Homoskedastic design | (c) Sample complexity as $\kappa$ increases |

Figure 1: We compare the sampling strategies of algorithms that do and do not account for heteroskedastic variance in Figs. 1a and 1b respectively, when $\kappa = 20$. Figure 1c shows that the gap in the optimal sample complexity between the two algorithms becomes linear as $\kappa$ increases.

have a multiplicative dependence on $\kappa$, we only suffer an *additive* dependence. Lastly, we show that the theoretical improvements translate to stronger empirical performance.

## 2   Related Work

**Experimental Design and Active Learning for Heteroskedastic Regression.** Many experimental design methods for heteroskedastic regression assume known weights (or efficiency functions) [55, 54]. Procedures that estimate the heteroskedastic variance focus on optimizing the design for the weighted least squares estimator given a burn-in period used to estimate the noise parameters [37, 53, 52]. These contributions assume that the variance is unstructured, which leads to sample complexities that scale with the number of arms. Similarly, past active learning work on linear bandits with heteroskedastic noise either assume the variances are known [30, 62], or that the variances are unstructured [14, 8, 4]. The works that assume structured heteroskedastic noise in a linear bandit framework focus on active maximum likelihood estimation and G-optimal design [7, 45], but do not exploit the structure of the heteroskedastic noise to improve performance, leading to complexities scaling poorly in the problem dimension. Lastly, Mukherjee et al. [38] study policy value estimation under the reward model described in Eq. 1. However, their variance estimator is different from ours.

**Best-Arm and Level Set Identification in Linear and Logistic Bandits.** Best-arm identification is a fundamental problem in the bandit literature [13, 46, 18]. Soare [44] constructed the first passive and adaptive algorithms for pure-exploration in linear bandits using G-optimal and $\mathcal{XY}$-allocation experimental designs. Fiez et al. [12] built on these results, developing the first pure exploration algorithm (RAGE) for the transductive linear bandit setting with nearly matching upper and lower bounds. Recent work specializes these methods to different settings [47, 57, 25, 26], improves time complexity [60, 50, 33], and targets different (minimax, asymptotically optimal, etc.) lower bound guarantees [59, 22, 9]. In the level set literature, Mason et al. [35] provides the first instance optimal algorithm with matching upper and lower bounds. Critically, none of these contributions account for heteroskedastic variance. Finally, we note the connection between this work and the logistic bandits literature [23, 11, 1, 36]. In that setting, the observations are Bernoulli random variables with a probability given by $\mathbb{P}(y = 1|x) = \left\{1 + \exp(-x^\top \theta^*)\right\}^{-1}$ for $x \in \mathbb{R}^d, \theta^* \in \mathbb{R}^d$ and $y \in \{0, 1\}$. Because of the mean-variance relationship for Bernoulli random variables, points $x$ for which $\mathbb{P}(y = 1|x)$ is near 0 or 1 have lower variance than other $x$'s. While the tools are different in these two cases, we highlight two common ideas: First, Mason et al. [36] present a method that explicitly handles the heteroskedasticity in logistic bandits. Second, a core focus of optimal logistic bandit papers is reducing the dependence on the worst case variance to only an additive penalty, similar to the guarantee of $\kappa$ herein, which has been shown to be unavoidable in general [23].

## 3   Heteroskedastic Variance Estimation

In this section, we describe an algorithm for heteroskedastic noise variance estimation with error bounds that scale favorably in the problem dimension. We then evaluate this method empirically.

Given a sampling budget $\Gamma$, our goal is to choose $\{x_t\}_{t=1}^{\Gamma} \subset \mathcal{X}$ to control the tail of the *maximum absolute error (MAE):* $\max_{x \in \mathcal{X}} |\widehat{\sigma}_x^2 - \sigma_x^2|$ by exploiting the heteroskedastic variance model. Alg. 1, HEAD (Heteroskedasticity Estimation by Adaptive Designs), provides an experimental design for finite-time bounds on structured heteroskedastic variance estimation.

**Notation.** Let $M := d(d+1)/2$. Define $\mathcal{W} = \{\phi_x, \forall x \in \mathcal{X}\} \subset \mathbb{R}^M$ where $\phi_x = \text{vec}(xx^\top)G$ and $G \in \mathbb{R}^{d^2 \times M}$ is the duplication matrix such that $G \text{vech}(xx^\top) = \text{vec}(xx^\top)$, with $\text{vech}(\cdot)$ denoting the half-vectorization. For a set $\mathcal{V}$, let $P_{\mathcal{V}} := \{\lambda \in \mathbb{R}^{|\mathcal{V}|} : \sum_{v \in \mathcal{V}} \lambda_v = 1, \lambda_v \geq 0 \ \forall \ v\}$ denote distributions over $\mathcal{V}$ and $\mathcal{Y}(\mathcal{V}) := \{v' - v : v, v' \in \mathcal{V}, \ v \neq v'\}$ represent the differences between members of set $\mathcal{V}$.

**General Approach.** Given observations from $y_t = x_t^\top \theta^* + \eta_t$ for $t \in \{1, 2, \ldots, \Gamma\}$ and the true value of $\theta^*$, we can estimate $\sigma_x^2$ for each $x \in \mathcal{X}$ from the squared residuals $\eta_t^2 = (y_t - x_t^\top \theta^*)^2$ (e.g., using method of moments or another estimating equation approach [15]). However, since $\theta^*$ is not known, we cannot directly observe $\eta_t^2$, making estimation of $\sigma_x^2$ more difficult. Consider a general procedure that first uses $\Gamma/2$ samples from the budget to construct a least-squares estimator for $\theta^*$, $\widehat{\theta}_{\Gamma/2}$, and then uses the final $\Gamma/2$ samples to collect error estimates of the form $\widehat{\eta}_t^2 = (y_t - x_t^\top \widehat{\theta}_{\Gamma/2})^2$. Define $\Phi_\Gamma \in \mathbb{R}^{\Gamma/2 \times M}$ as a matrix with rows $\{\phi_{x_t}\}_{t=\Gamma/2+1}^{\Gamma}$ and $\widehat{\eta} \in \mathbb{R}^{\Gamma/2}$ as a vector with values $\{\widehat{\eta}_t\}_{t=\Gamma/2+1}^{\Gamma}$. With $\widehat{\eta}$, we can estimate $\Sigma^*$ by least squares as

$$\text{vech}(\widehat{\Sigma}_\Gamma) = \underset{\mathbf{s} \in \mathbb{R}^M}{\arg\min} \|\Phi_\Gamma \mathbf{s} - \widehat{\eta}^2\|^2,$$

where we minimize over the space of symmetric matrices. Splitting samples ensures that the errors in estimating $\theta^*$ and $\Sigma^*$ are independent. Note that $(\widehat{\Sigma}_\Gamma, \widehat{\theta}_{\Gamma/2})$ is an M-estimator of $(\Sigma^*, \theta^*)$ derived from unbiased estimating functions, a common approach in heteroskedastic regression [21, 51, 34]. Finally, by plugging in the estimate of $\Sigma^*$, one can obtain an estimate of the variance, $\widehat{\sigma}_x^2 = \min\{\max\{x^\top \widehat{\Sigma}_\Gamma x, \sigma_{\min}^2\}, \sigma_{\max}^2\}$. We show that this approach leads to the following general error decomposition for any $x \in \mathcal{X}$ with $z := \phi_x^\top (\Phi_\Gamma^\top \Phi_\Gamma)^{-1} \Phi_\Gamma^\top$:

$$|x^\top (\Sigma^* - \widehat{\Sigma}_\Gamma)x| \leq \underbrace{\left\{ x^\top (\widehat{\theta}_{\Gamma/2} - \theta^*) \right\}^2}_{\mathbf{A}} + \underbrace{\left| \sum_{t=\Gamma/2}^{\Gamma} z_t \left( \eta_t^2 - x_t^\top \Sigma^* x_t \right) \right|}_{\mathbf{B}} + \underbrace{\left| \sum_{t=\Gamma/2}^{\Gamma} 2 z_t x_t^\top (\widehat{\theta}_{\Gamma/2} - \theta^*) \eta_t \right|}_{\mathbf{C}}.$$

**Experimental Design for Precise Estimation.** The analysis above gives an error decomposition for any phased non-adaptive data collection method that splits samples between estimation of $\theta^*$ and $\Sigma^*$. Intuitively, we wish to sample $\{x_1, \ldots, x_{\Gamma/2}\}$ and $\{x_{\Gamma/2+1}, \ldots, x_\Gamma\}$ to control the maximum absolute error by minimizing the expression above. We achieve this via two phases of G-optimal experimental design [28]– first over the $\mathcal{X}$ space to estimate $\theta^*$, and then over the $\mathcal{W}$ space to estimate $\Sigma^*$. Precisely, we control quantity **A** via stage 1 of Alg. 1, which draws from a G-optimal design $\lambda^*$ to minimize the maximum predictive variance

$$\max_{x \in \mathcal{X}} \mathbb{E} \left[ \left\{ x^\top (\widehat{\theta}_{\Gamma/2} - \theta^*) \right\}^2 \right].$$

Quantity **B** is another sub-exponential quantity (Definition B.2 in Appendix B), which we control by drawing from a G-optimal design $\lambda^\dagger$ over $\mathcal{W}$ to minimize $\max_{\phi \in \mathcal{W}} \phi^\top (\Phi_\Gamma^\top \Phi_\Gamma)^{-1} \phi$. Finally, **C** is controlled by a combination of the guarantees associated with $\lambda^*$ and $\lambda^\dagger$. This leads to the following error bound on $\widehat{\sigma}_x^2 = \min \left\{ \max\{x^\top \widehat{\Sigma}_\Gamma x, \sigma_{\min}^2\}, \sigma_{\max}^2 \right\}$.

**Theorem 3.1.** Assume $\Gamma = \Omega \left[ \max \left\{ \sigma_{\max}^2 \log(|\mathcal{X}|/\delta)d^2, d^2 \right\} \right]$. For any $x \in \mathcal{X}$ and $\delta \in (0, 1)$, Alg. 1 guarantees the following:

$$\mathbb{P} \left( |\widehat{\sigma}_x^2 - \sigma_x^2| \leq C_{\Gamma, \delta} \right) \geq 1 - \delta/2 \quad \text{where} \quad C_{\Gamma, \delta} = \mathcal{O} \left\{ \sqrt{\log(|\mathcal{X}|/\delta) \sigma_{\max}^2 d^2 / \Gamma} \right\}.$$

The proof of Theorem 3.1 is in Appendix D and follows from the decomposition in Eq. 3; treating **A**, **B** and **C** as sub-exponential random variables with bounds that leverage the experimental designs of Alg. 1. The details on the constants involved in Theorem 3.1 are also included in Appendix D.

---

**Algorithm 1:** HEAD (Heteroskedasticity Estimation by Adaptive Designs)

    **Result:** Find $\widehat{\Sigma}_\Gamma$

**1** **Input:** Arms $\mathcal{X} \in \mathbb{R}^d, \Gamma \in \mathbb{N}$

**2** //Stage 1:  Take half the samples to estimate $\theta^*$

**3** Determine $\lambda^*$ according to $\lambda^* = \arg\min_{\lambda \in P_\mathcal{X}} \max_{x \in \mathcal{X}} x^\top \left( \sum_{x' \in \mathcal{X}} \lambda_{x'} x' x'^\top \right)^{-1} x$

**4** Pull arm $x \in \mathcal{X} \lceil \lambda_x^* \Gamma/2 \rceil$ times and collect observations $\{x_t, y_t\}_{t=1}^{\Gamma/2}$

**5** Define $A^* = \sum_{t=1}^{\Gamma/2} x_t x_t^\top$ and $b^* = \sum_{t=1}^{\Gamma/2} x_t y_t$ and estimate $\widehat{\theta}_{\Gamma/2} = A^{*-1} b$

**6** //Stage 2:  Take half the samples to estimate $\Sigma^*$ given $\widehat{\theta}_{\Gamma/2}$

**7** Determine $\lambda^\dagger \leftarrow \lambda_x^\dagger = \arg\min_{\lambda \in P_\mathcal{X}} \max_{x \in \mathcal{X}} \phi_x^\top \left( \sum_{x' \in \mathcal{X}} \lambda_{x'} \phi_{x'} \phi_{x'}^\top \right)^{-1} \phi_x$

**8** Pull arm $x \in \mathcal{X} \lceil \lambda_x^\dagger \Gamma/2 \rceil$ times[8] and collect observations $\{x_t, y_t\}_{t=\Gamma/2+1}^{\Gamma}$

**9** Let $A^\dagger = \sum_{t=\Gamma/2+1}^{\Gamma} \phi_{x_t} \phi_{x_t}^\top, b^\dagger = \sum_{t=\Gamma/2+1}^{\Gamma} \phi_{x_t} \left( y_t - x_t^\top \widehat{\theta}_{\Gamma/2} \right)^2$

**10** **Output:** $\operatorname{vech}(\widehat{\Sigma}_\Gamma) = A^{\dagger -1} b^\dagger$.

---

Note that we design to directly control the MAE of $\widehat{\sigma}_x^2$. This is in contrast to an inefficient alternative procedure that allocates samples to minimize the error of $\widehat{\Sigma}_\Gamma$ in general, and then extends this bound to $x^\top \widehat{\Sigma}_\Gamma x$ for $x \in \mathcal{X}$.

**Theoretical Comparison of Variance Estimation Methods.** In contrast to previous approaches that naively scale in the size of $\mathcal{X}$ [14], the above result has a dimension dependence of $d^2$, which intuitively scales with the degrees of freedom of $\Sigma^*$. To highlight the tightness of this result, consider estimating the noise parameters for each arm, $x \in \mathcal{X}$, only using the samples of $x$ (like the method in [14]). We improve this approach by adapting it to our heteroskedastic variance structure and strategically pick $d^2$ points to avoid the dependence on $|\mathcal{X}|$. We refer to this method as the *Separate Arm Estimator*, and in Appendix D, we show that it suffers a dependence of $d^4$. This comparison point, along with our empirical results below, suggest that the error bounds established in Theorem 3.1 scale favorably with the problem dimension.

**Empirical Comparison of Variance Estimation Methods.** Define two sets of arms $\mathcal{X}_1, \mathcal{X}_2$ such that $x \in \mathcal{X}_1$ is drawn uniformly from a unit sphere and $x \in \mathcal{X}_2$ is drawn uniformly from a sphere with radius 0.1; with $|\mathcal{X}_1| = 200$ and $|\mathcal{X}_2| = 1800$. Let $\Sigma^* = \operatorname{diag}(1, 0.1, 1, 0.1, \dots) \in \mathbb{R}^{d \times d}$, and $\theta^* = (1, 1, \dots, 1) \in \mathbb{R}^d$. This setting provides the heteroskedastic variation needed to illustrate the advantages of using HEAD (Alg. 1). An optimal algorithm will tend to target orthogonal vectors in $\mathcal{X}_1$ because these arms have higher magnitudes in informative directions. Defining $\mathcal{X} = \mathcal{X}_1 \cup \mathcal{X}_2$, we perform 32 simulations in which we randomly sample the arm set and construct estimators for each $\sigma_x^2$, $x \in \mathcal{X}$. We compare HEAD, the Separate Arm Estimator, and the Uniform Estimator (see Alg. 3 in Appendix D). The Uniform Estimator is based on Alg. 1 but samples arms uniformly from $\mathcal{X}$ and does not split samples between estimation of $\theta^*$ and $\Sigma^*$. HEAD should outperform its competitors in two ways: 1) optimally allocating samples, and 2) efficiently sharing information across arms in estimation. The Uniform Estimator exploits the problem structure for estimation but not when sampling. In contrast, the Seperate Arm Estimator optimally samples, but does not use the relationships between arms for estimation. Fig. 2 depicts the average MAE and standard errors for each estimator. In Fig. 2a, we see that HEAD outperforms its competitors over a series of sample sizes for $d = 15$. Fig. 2b compares the estimators over a range of dimensions for a sample size of $95k$. While HEAD continues to accurately estimate the heteroskedastic noise parameters at high dimensions, the Separate Arm Estimator error scales poorly with dimension as the analysis suggests.

## 4 Adaptive Experimentation with Covariates in Heteroskedastic Settings

As an application of the novel sampling procedure and resulting variance estimation bounds developed in section 3, we now study *adaptive experimentation with covariates* in the presence of heteroskedastic

---

[8]As $\lambda_x \Gamma/2 \notin \mathbb{N}$, it is not possible in reality to take $\lambda_x \Gamma/2$ samples. This problem can easily be handled by rounding procedures detailed in Appendix A.

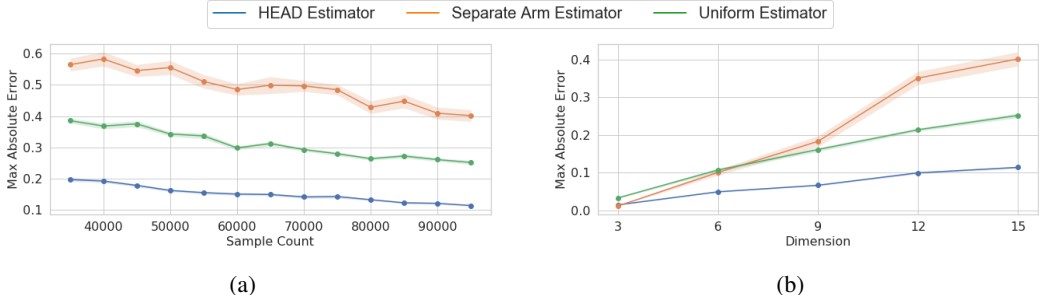

Figure 2: The proposed experimental design estimator outperforms competitors in terms of maximum absolute error $\max_{x \in \mathcal{X}} |\sigma_x^2 - \widehat{\sigma}_x^2|$ over a range of sample sizes (a) and dimensions (b).

noise through the lens of *transductive linear identification* problems [12]. Recalling the problem setting from Section 1.1, we consider the following fixed confidence identification objectives:

1. *Best-Arm Identification* (BAI). Identify the singleton set $\{z^*\}$ where $z^* = \arg\max_{z \in \mathcal{Z}} z^\top \theta^*$.

2. *Level-Set Identification* (LS). Identify the set $G_\alpha = \{z : z^\top \theta^* > \alpha\}$ given a threshold $\alpha \in \mathbb{R}$.

This set of objectives, together with a learning protocol, allow us to capture a number of practical problems of interest where heteroskedastic noise naturally arises in the assumed form. Consider multivariate testing problems in e-commerce advertisement applications [17, 39]. The expected feedback is modeled through a linear combination of primary effects from content variation dimensions (e.g., features or locations that partition the advertisement), and interaction effects between content variation in pairs of dimensions. The noise structure is naturally heteroskedastic and dependent on the combination of variation choices. Moreover, the transductive objectives allow for flexible experimentation such as identifying the optimal combination of variations or the primary effects that exceed some threshold. We return to this setting in Experiment 3 of Section 5. In the remainder of this section, we characterize the optimal sample complexities for the heteroskedastic transductive linear identification problems and design algorithms that nearly match these lower bounds.

## 4.1 Linear Estimation Methods with Heteroskedastic Noise

To draw inferences in the experimentation problems of interest, it is necessary to consider estimation methods for the unknown parameter vector $\theta^* \in \mathbb{R}^d$. Given a matrix of covariates $X_T \in \mathbb{R}^{T \times d}$, a vector of observations $Y_T \in \mathbb{R}^T$, and a user-defined weighting matrix $W_T = \mathrm{diag}(w_1, \ldots, w_T) \in \mathbb{R}^{T \times T}$, the weighted least squares (WLS) estimator is defined as:

$$\widehat{\theta}_{\mathrm{WLS}} = \left(X_T^\top W_T X_T\right)^{-1} X_T^\top W_T Y_T = \min_{\theta \in \mathbb{R}^d} \|Y_T - X_T \theta\|_{W_T}^2.$$

Let $\Sigma_T = \mathrm{diag}(\sigma_{x_1}^2, \sigma_{x_2}^2, \ldots, \sigma_{x_T}^2) \in \mathbb{R}^{T \times T}$ denote the variance parameters of the noise observations. The WLS estimator is unbiased regardless of the weight selections, with variance

$$\mathbb{V}\left(\widehat{\theta}_{\mathrm{WLS}}\right) = \left(X_T^\top W_T X_T\right)^{-1} X_T^\top W_T \Sigma_T W_T^\top X_T \left(X_T^\top W_T X_T\right)^{-1} \overset{W_T \to \Sigma_T^{-1}}{=} \left(X_T^\top \Sigma_T^{-1} X_T\right)^{-1}.$$
(2)

The WLS estimator with $W_T = \Sigma_T^{-1}$ is the minimum variance linear unbiased estimator [2]. In contrast, the ordinary least squares (OLS) estimator, obtained by taking $W_T = I$ in the WLS estimator, is unbiased but with a higher variance,

$$\mathbb{V}\left(\widehat{\theta}_{\mathrm{OLS}}\right) = \left(X_T^\top X_T\right)^{-1} X_T^\top \Sigma_T X_T \left(X_T^\top X_T\right)^{-1} \preceq \sigma_{\max}^2 (X_T^\top X_T)^{-1}.$$

Consequently, WLS is a natural choice with heteroskedastic noise for sample efficient estimation.

## 4.2 Optimal Sample Complexities

We now motivate the optimal sample complexity for any algorithm that returns the correct quantity with a fixed probability greater than $1 - \delta$, $\delta \in (0, 1)$, in transductive identification problems with

heteroskedastic noise. We seek to identify a set of vectors $\mathcal{H}_{\mathrm{OBJ}}$, where

$$\mathcal{H}_{\mathrm{BAI}} = \{z^*\} = \{\arg\max_{z \in \mathcal{Z}} z^\top \theta^*\} \text{ and } \mathcal{H}_{\mathrm{LS}} = \{z \in \mathcal{Z} : z^\top \theta^* - \alpha > 0\}.$$

Algorithms consist of a sampling strategy and a rule to return a set $\widehat{\mathcal{H}}$ at time $\tau$.

**Definition 4.1.** An algorithm is $\delta$-PAC for $\mathrm{OBJ} \in \{\mathrm{BAI}, \mathrm{LS}\}$ if $\forall \theta \in \Theta$, $\mathbb{P}_\theta(\widehat{\mathcal{H}} = \mathcal{H}_{\mathrm{OBJ}}) \geq 1 - \delta$.

Throughout, we will only consider algorithms that are $\delta$-PAC. For drawing parallels between best-arm and level set identification, it will be useful to define a set of values $\mathcal{Q}_{\mathrm{OBJ}}$, where

$$\mathcal{Q}_{\mathrm{BAI}} = \{z : z \in \mathcal{Z}, z \neq z^*\} \text{ and } \mathcal{Q}_{\mathrm{LS}} = \{0\},$$

and a constant $b_{\mathrm{OBJ}}$ such that $b_{\mathrm{BAI}} = 0$ and $b_{\mathrm{LS}} = \alpha$. Both problems amount to verifying that for all $h \in \mathcal{H}_{\mathrm{OBJ}}$ and $q \in \mathcal{Q}_{\mathrm{OBJ}}$, $(h - q)^\top \widehat{\theta}_{\mathrm{WLS}} > b_{\mathrm{OBJ}}$, or equivalently $(h - q)^\top \theta^* - b_{\mathrm{OBJ}} - (h - q)^\top (\theta^* - \widehat{\theta}_{\mathrm{WLS}}) > 0$. Using a sub-Gaussian tail bound and a union bound [32], this verification requires the following to hold for all $h \in \mathcal{H}_{\mathrm{OBJ}}$ and $q \in \mathcal{Q}_{\mathrm{OBJ}}$ for the data collected:

$$|(h - q)^\top (\theta^* - \widehat{\theta}_{\mathrm{WLS}})| \leq \sqrt{2\|h - q\|^2_{\mathbb{V}(\widehat{\theta}_{\mathrm{WLS}})} \log(2|\mathcal{Z}|/\delta)} \leq (h - q)^\top \theta^* - b_{\mathrm{OBJ}}. \quad (3)$$

Let $\lambda_x$ denote the proportion of $T$ samples given to a measurement vector $x \in \mathcal{X}$ so that when $W_T = \Sigma_T^{-1}$ in Eq. (4.1),

$$\mathbb{V}\left(\widehat{\theta}_{\mathrm{WLS}}\right) = \left(X_T^\top \Sigma_T^{-1} X_T\right)^{-1} = \left(\sum_{x \in \mathcal{X}} \lambda_x \sigma_x^{-2} xx^\top\right)^{-1} / T$$

as in Eq. (2). Now, we reformulate the condition in Eq. (3) and minimize over designs $\lambda \in P_\mathcal{X}$ to see that $\delta$-PAC verification requires

$$T \geq 2\psi^*_{\mathrm{OBJ}} \log(2|\mathcal{Z}|/\delta), \text{ where } \psi^*_{\mathrm{OBJ}} = \min_{\lambda \in P_\mathcal{X}} \max_{h \in \mathcal{H}_{\mathrm{OBJ}}, q \in \mathcal{Q}_{\mathrm{OBJ}}} \frac{\|h - q\|^2_{(\sum_{x \in \mathcal{X}} \lambda_x \sigma_x^{-2} xx^\top)^{-1}}}{\{(h - q)^\top \theta^* - b_{\mathrm{OBJ}}\}^2}. \quad (4)$$

This motivating analysis gives rise to the following sample complexity lower bound.

**Theorem 4.1.** Consider an objective, OBJ, of best-arm identification (BAI) or level-set identification (LS). For any $\delta \in (0, 1)$, a $\delta$-PAC algorithm must satisfy

$$E_\theta[\tau] \geq 2 \log(1/2.4\delta)\psi^*_{\mathrm{OBJ}}.$$

**Remark 4.1.** This lower bound assumes that the noise variance parameters are *known*. We compare to the existing lower bounds for the identification objectives with homoskedastic noise [12, 44, 35] in Appendix G. These apply in this setting by taking the variance parameter $\sigma^2 = \sigma^2_{\mathrm{max}}$ and are characterized by an instance-dependent parameter, $\rho^*_{\mathrm{OBJ}}$, such that $\kappa^{-1} \leq \psi^*_{\mathrm{OBJ}}/\rho^*_{\mathrm{OBJ}} \leq 1$ (recall that $\kappa := \sigma^2_{\mathrm{max}}/\sigma^2_{\mathrm{min}}$). In general, $\kappa$ can be quite large in many problems with heteroskedastic noise. Consequently, this implies that near-optimal algorithms for linear identification with homoskedastic noise can be *highly suboptimal* (by up to a multiplicative factor of $\kappa$) in this setting.

## 4.3 Adaptive Designs with Unknown Heteroskedastic Variance

We now present Alg. 2, `H-RAGE` (Heteroskedastic Randomized Adaptive Gap Elimination), as a solution for adaptive experimentation with covariates in heteroskedastic settings. The approach is composed of two components: 1) Obtain accurate estimates of the unknown variance parameters $\{\sigma_x^2\}_{x \in \mathcal{X}}$ using `HEAD`; 2) Use adaptive experimental design methods for minimizing the uncertainty in directions of interest given the estimated variance parameters. Denote by $\Delta = \min_{h \in \mathcal{H}_{\mathrm{OBJ}}} \min_{q \in \mathcal{Q}_{\mathrm{OBJ}}} (h - q)^\top \theta^* - b_{\mathrm{OBJ}}$ the minimum gap for the objective and assume $\max_{z \in \mathcal{Z}_\ell} |\langle z - z^*, \theta^* \rangle| \leq 2$. We prove the following guarantees for Alg. 2.

**Theorem 4.2.** Consider an objective, OBJ, of best-arm identification (BAI) or level-set identification (LS). The set returned from Alg. 2 at a time $\tau$ is $\delta$-PAC and

$$\tau = \mathcal{O}\left(\psi^*_{\mathrm{OBJ}} \log(\Delta^{-1}) \left[\log(|\mathcal{Z}|/\delta) + \log\{\log(\Delta^{-1})\}\right] + \log(\Delta^{-1})d^2 + \log(|\mathcal{X}|/\delta)\kappa^2 d^2\right).$$

---

**Algorithm 2:** (H-RAGE) Heteroskedastic Randomized Adaptive Gap Elimination

---

**Result:** Find $z^* := \arg\max_{z \in \mathcal{Z}} z^\top \theta^*$ for BAI or $G_\alpha := \{z \in \mathcal{Z} : z^\top \theta^* > \alpha\}$ for LS

1   **Input:** $\mathcal{X} \in \mathbb{R}^d$, $\mathcal{Z} \in \mathbb{R}^d$, confidence $\delta \in (0,1)$, OBJ $\in$ {BAI,LS}, threshold $\alpha \in \mathbb{R}$

2   **Initialize:** $\ell \leftarrow 1$, $\mathcal{Z}_1 \leftarrow \mathcal{Z}$, $\mathcal{G}_1 \leftarrow \emptyset$, $\mathcal{B}_1 \leftarrow \emptyset$

3   //Variance estimation

4   Call Alg. 1 such that $\left| \widehat{\sigma}_x^2 = \min\left\{ \max\{x^\top \widehat{\Sigma}_\Gamma x, \sigma_{\min}^2\}, \sigma_{\max}^2 \right\} - \sigma_x^2 \right| \leq \sigma_x^2/2$

5   **while** $(|\mathcal{Z}_\ell| > 1$ *and* OBJ=BAI$)$ *or* $(|\mathcal{Z}_\ell| > 0$ *and* OBJ=LS$)$ **do**

6      //Determine the design

7      Let $\widehat{\lambda}_\ell \in P_\mathcal{X}$ be a minimizer of $q\{\lambda, \mathcal{Y}(\mathcal{Z}_\ell)\}$ if OBJ=BAI and $q(\lambda, \mathcal{Z}_\ell)$ if OBJ=LS where

$$q(\mathcal{V}) = \inf_{\lambda \in P_\mathcal{X}} q(\lambda; \mathcal{V}) = \inf_{\lambda \in P_\mathcal{X}} \max_{z \in \mathcal{V}} ||z||^2_{(\sum_{x \in \mathcal{X}} \widehat{\sigma}_x^{-2} \lambda_x x x^\top)^{-1}}$$

8      Set $\epsilon_\ell = 2^{-\ell}$, $\tau_\ell = 3\epsilon_\ell^{-2} q(\mathcal{Z}_\ell) \log(8\ell^2 |\mathcal{Z}|/\delta)$ //Determine stepsize

9      Pull arm $x \in \mathcal{X}$ exactly $\lceil \tau_\ell \widehat{\lambda}_{\ell,x} \rceil$ times for $n_\ell$ samples and collect $\{x_{\ell,i}, y_{\ell,i}\}_{i=1}^{n_\ell}$

10     Define $A_\ell := \sum_{i=1}^{n_\ell} \widehat{\sigma}_i^{-2} x_{\ell,i} x_{\ell,i}^\top$, $b_\ell = \sum_{i=1}^{n_\ell} \widehat{\sigma}_i^{-2} x_{\ell,i} y_{\ell,i}$ and construct $\widehat{\theta}_\ell = A_\ell^{-1} b_\ell$

11     //Eliminate arms

12     **if** *OBJ is BAI* **then**

13        $\mathcal{Z}_{\ell+1} \leftarrow \mathcal{Z}_\ell \setminus \{z \in \mathcal{Z}_\ell : \max_{z' \in \mathcal{Z}_\ell} \langle z' - z, \widehat{\theta}_\ell \rangle > \epsilon_\ell\}$

14     **else**

15        $\mathcal{G}_{\ell+1} \leftarrow \mathcal{G}_\ell \cup \{z \in \mathcal{Z}_\ell : \langle z, \widehat{\theta}_\ell \rangle - \epsilon_\ell > \alpha\}$

16        $\mathcal{B}_{\ell+1} \leftarrow \mathcal{B}_\ell \cup \{z \in \mathcal{Z}_\ell : \langle z, \widehat{\theta}_\ell \rangle + \epsilon_\ell < \alpha\}$

17        $\mathcal{Z}_{\ell+1} \leftarrow \mathcal{Z}_\ell \setminus \{\{\mathcal{G}_\ell \setminus \mathcal{G}_{\ell+1}\} \cup \{\mathcal{B}_\ell \setminus \mathcal{B}_{\ell+1}\}\}$

18     **end**

19     $\ell \leftarrow \ell + 1$

20   **end**

21   **Output:** $\mathcal{Z}_\ell$ for BAI or $\mathcal{G}_\ell$ for LS

---

**Remark 4.2.** This result shows that Alg. 2 is nearly instance-optimal. Critically, the impact of not knowing the noise variances ahead of time only has an *additive* impact on the sample complexity relative to the lower bound. In contrast, existing near-optimal methods for identification problems with homoskedastic noise would be suboptimal by a *multiplicative* factor depending on $\kappa$ relative to the lower bound. Given that the lower bound assumes the variances are known, an interesting question for future work is a tighter lower bound assuming the variance parameters are unknown.

**Algorithm Overview.** Observe that for $\psi^*_{\text{OBJ}}$, the optimal allocation depends on both the unknown gaps $((h - q)^\top \theta^* - b_{\text{OBJ}})$ for $h \in \mathcal{H}_{\text{OBJ}}$ and $q \in \mathcal{Q}_{\text{OBJ}}$, and the unknown noise variance parameters $\{\sigma_x^2\}_{x \in \mathcal{X}}$. To handle this, the algorithm begins with an initial burn-in phase using the procedure from Alg. 1 to produce estimates $\{\widehat{\sigma}_x^2\}_{x \in \mathcal{X}}$ of the unknown parameters $\{\sigma_x^2\}_{x \in \mathcal{X}}$, achieving a multiplicative error bound of the form $|\sigma_x^2 - \widehat{\sigma}_x^2| \leq \sigma_x^2/2$ for all $x \in \mathcal{X}$. From this point, the estimates $\{\widehat{\sigma}_x^2\}_{x \in \mathcal{X}}$ are used as plug-ins to the experimental designs and the weighted least squares estimates. In each round $\ell \in \mathbb{N}$ of the procedure, we maintain a set of uncertain items $\mathcal{Z}_\ell$, and obtain the sampling distribution $\widehat{\lambda}_\ell$, by minimizing the maximum predictive variance of the WLS estimator over all directions $y = h - q$ for $h \in \mathcal{H}_{\text{OBJ}}$ and $q \in \mathcal{Q}_{\text{OBJ}}$. This is known as an $\mathcal{X}\mathcal{Y}$-allocation and $G$-optimal-allocation in the best-arm and level set identification problems respectively. Critically, the number of samples taken in round $\ell$ guarantees that the error in all estimates is bounded by $\epsilon_\ell$, such that for any $h \in \mathcal{H}_{\text{OBJ}}$ and $q \in \mathcal{Q}_{\text{OBJ}}$, $(h - q)^\top \theta^* > 2\epsilon_\ell$ is eliminated from the active set at the end of the round. By progressively honing in on the optimal item set $\mathcal{H}_{\text{OBJ}}$, the sampling allocation gets closer to approximating the oracle allocation.

## 5   Experiments

We now present experiments focusing on the transductive linear bandit setting. We compare H-RAGE and its corresponding oracle to their homoskedastic counterparts, RAGE [12] and the oracle allocation

with noise equal to $\sigma^2_{\max}$. RAGE is provably near-optimal in the homoskedastic setting and known to perform well empirically. The algorithm is similar in concept to Alg. 2, but lacks the initial exploratory phase to estimate the variance parameters. We include the pseudo-code of RAGE in Appendix H. The homoskedastic and heteroskedastic oracles sample with respect to the optimal design with a sub-Gaussian interval stopping condition. All algorithms are run at a confidence level of $\delta = 0.05$, and we use the Franke-Wolfe method to compute the designs. We demonstrate that H-RAGE has superior empirical performance over a range of settings.

**Linear bandits: Experiment 1 (Signal-to-Noise).** We begin by presenting an experiment that is a variation of the standard benchmark problem for BAI in linear bandits [46] introduced in Sec. 1.2. Define $e_i \in \mathbb{R}^d$ as the standard basis vectors in $d$ dimensions. In the standard example, $\mathcal{X} = \mathcal{Z} = \{e_1, e_2, x'\}$ where $x' = \{e_1 \cos(\omega) + e_2 \sin(\omega)\}$ for $\omega \to 0$, and $\theta^* = e_1$. Thus $x_1 = e_1$ is optimal, $x_3 = x'$ is near-optimal, and $x_2 = e_2$ is informative to discriminate between $x_1$ and $x_3$. We consider an extension of this benchmark in multiple dimensions that highlights the performance gains from accounting for heteroskedastic variance. First, define an arm set $\mathcal{H}$ that is comprised of the standard example but with varied magnitudes. For $q \in (0, 1)$,

$$\mathcal{H} = \{e_1\} \cup \{e_2\} \cup \{q \times e_i\}_{i=3}^d \cup \{e_1 \cos(\omega) + e_i \sin(\omega)\}_{i=2}^d.$$

Then define $\mathcal{G}$ to be a series of vectors such that the rank of $\mathrm{span}(\{\phi_x : x \in \mathcal{G} \cup \mathcal{H}\}) = \mathbb{R}^M$. $\mathcal{G}$ allows for variance estimation. Finally, let the arm and item set be $\mathcal{X} = \mathcal{Z} = \mathcal{G} \cup \mathcal{H}$ and $\theta^* = e_1$.

We define $\Sigma^*$ as the $d$-dimensional identity matrix. Observe that $x_1 = e_1$ is optimal, the $d - 1$ arms given by $\{e_1 \cos(\omega) + e_i \sin(\omega)\}_{i=2}^d$ are near-optimal with equal gaps $\Delta = 1 - \cos(\omega)$, and the $d - 1$ arms given by $\{e_2\} \cup q * \{e_i\}_{i=3}^d$ are informative to sample for discriminating between $x_1 = e_1$ and $x_i = e_1 \cos(\omega) + e_i \sin(\omega)$ for each $i \in \{2, \ldots, d\}$, respectively. To identify $x_1$, we must estimate the $\Delta$ gap adequately in every dimension with arms $\{e_2\} \cup q * \{e_i\}_{i=3}^d$. Therefore, performing BAI without accounting for heteroskedastic variances will tend toward a design that prioritizes sampling informative arms $\{q \times e_i\}_{i=3}^d$, since these have smaller magnitudes and seem less informative about the near-optimal arms in dimensions $i \in \{3, 4 \ldots d\}$. However, the signal to noise ratio is actually constant across arms (because the variance of $q * \{e_i\}_{i=3}^d$ is equal to $q^2$) leading to an oracle distribution that equally samples across $\{e_2\} \cup q * \{e_i\}_{i=3}^d$ when we account for heteroskedastic variances. This change in allocation is illustrated in Appendix H for $d = 4$, $\omega = 0.01$, $q = 0.4$ and $\mathcal{G} = \{0.5(e_1 + e_2) + 0.1e_3, 0.5(e_1 + e_2) + 0.1(e_3 + e_4)\}$. In the same setting, Fig. 3a shows that the heteroskedastic variance experimental design and the tighter predictive confidence intervals of the weighted least squares estimator result in a large decrease in the sample size needed to identify the best arm.

**Experiment 2: Benchmark.** This experiment is also similar to the benchmark example introduced in Sec. 1.2. In this variation, the informative arms have the same magnitude but are bent at a $45°$ angle,

$$\mathcal{X} = \{e_1\} \cup \{e_1 \cos(\omega) + e_2 \sin(\omega)\} \cup \{e_i\}_{i=3}^d \cup \{\{e_i/\sqrt{2} + e_j/\sqrt{2}\}_{i=1}^d\}_{j>i}.$$

Let the arm and item set be $\mathcal{X} = \mathcal{Z}$ and $\theta^* = e_1$. In Experiment 1, we have many near-optimal arms and H-RAGE reveals that each of these are equally difficult to distinguish from the best. In contrast, Experiment 2 contains one near-optimal arm along with many potentially informative arms, and we are interested in identifying the one that is most helpful. Intuitively, Experiment 1 uses heteroskedastic variance estimates to assess the difficulty of many problems, while Experiment 2 uses the same information to assess the benefits of many solutions. Let the unknown noise matrix be given by $\Sigma^* = \mathrm{diag}(\sigma_1^2, \sigma_2^2, \ldots, \sigma_d^2)$ where $\sigma_1^2 = \alpha^2$, $(\sigma_2^2, \sigma_3^2) = \beta^2$ and $(\sigma_4^2, \sigma_5^2 \ldots, \sigma_d^2) = \alpha^2$. $x_1 = e_1$ is again optimal and $x_2 = \{e_1 \cos(\omega) + e_2 \sin(\omega)\}$ is a near optimal arm. In this case, $\{e_2/\sqrt{2} + e_1/\sqrt{2}\} \cup \{e_2/\sqrt{2} + e_i/\sqrt{2}\}_{i=3}^d$ are informative for distinguishing between $x_1$ and $x_2$, and the oracle allocation assuming homoskedastic noise (Homoskedastic Oracle) picks three of these vectors to sample. However, if $\alpha^2 \gg \beta^2$, then it is optimal to sample $\{e_2/\sqrt{2} + e_3/\sqrt{2}, e_3\}$ over other potential informative arm combinations. Appendix H shows this contrast in allocations for $d = 3$, $\omega = 0.01$, $\alpha^2 = 1$ and $\beta^2 = 0.2$. In the same setting, Fig. 3b shows that estimating and accounting for heteroskedastic noise contributes to a reduction in sample complexity.

**Experiment 3: Multivariate Testing.** In this experiment, we return to the motivating example of Section 4, multivariate testing in e-commerce advertising [17, 39]. We divide an advertisement into natural locations or features called dimensions, each of which has different content options or variations. We induce heteroskedastic variance by allowing both the expected value and the variance

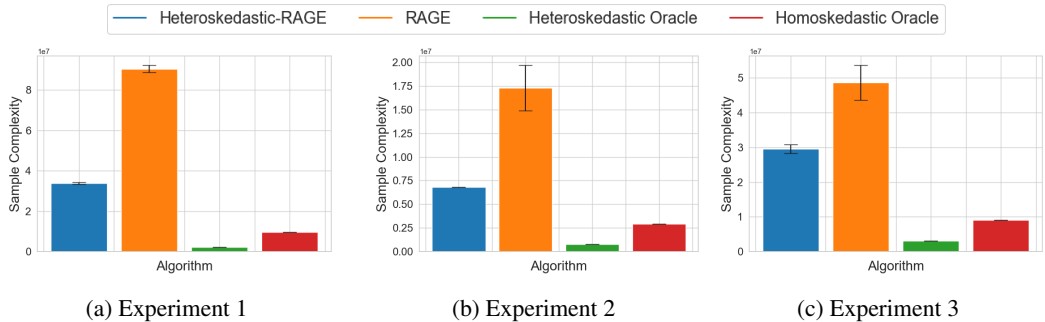

(a) Experiment 1        (b) Experiment 2        (c) Experiment 3

Figure 3: Experimental results show the mean sample complexity of each algorithm over 100 simulations with 95% Monte Carlo confidence intervals.

of an advertisement's reward to depend on the $n$ variations in each of the $m$ dimensions. Let $\mathcal{X} = \mathcal{Z}$ denote the set of layouts corresponding to the combinations of variant choices in the dimensions so that $|\mathcal{X}| = n^m$. The multivariate testing problem is often modeled in the linear bandit framework using the expected feedback for any layout $x \in \mathcal{X} \subset \{0, 1\}^d$, where $d = 1 + mn + n^2 m(m+1)/2$, given by

$$x^\top \theta^* := \theta_0^* + \left( \sum_{i=1}^m \sum_{j=1}^n \theta_{i,j}^* x_{i,j} \right) + \left( \sum_{i=1}^m \sum_{i'=i+1}^m \sum_{j=1}^n \sum_{j'=1}^n \theta_{i,i',j,j'}^* x_{i,j} x_{i',j'} \right).$$

In this model, $x_{i,j} \in \{0, 1\}$ is an indicator for variation $j \in \{1, \dots, n\}$ being placed in dimension $i \in \{1, \dots, m\}$, and $\sum_{j=1}^n x_{i,j} = 1$ for all $i \in \{1, \dots, n\}$ for any layout $x \in \mathcal{X}$. Observe that the expected feedback for a layout is modeled as a linear combination of a common bias term, primary effects from the variation choices within each dimension, and secondary interaction effects from the combinations of variation choices between dimensions.

We consider an environment where the effect of variation changes in one dimension, call it dimension $j$, has much higher variance than others; resulting in the best variation for dimension $j$ being harder to identify. An algorithm that accounts for heteroskedastic variance will devote a greater number of samples to compare variations in dimension $j$, whereas an algorithm that upper bounds the variance will split samples equally between dimensions. For a simulation setting with 2 variations in 3 dimensions, we define $\Sigma^* = \text{diag}(0.3, 0.7, 10^{-3}, 10^{-3}, \dots) \in \mathbb{R}^{7 \times 7}$ and $\theta^* = (0, 0.005, 0.0075, 0.01, -0.1, -0.1, \dots) \in \mathbb{R}^7$. This simulation setting implies that the second variation in each of the dimensions is better than the first, but the positive effect in the first dimension is hardest to identify. In Appendix H we can see that this causes the heteroskedastic oracle to devote more weight than the homoskedastic oracle to vectors that include the second variation in the first dimension. Fig. 3c depicts the improvement in sample complexity resulting from accounting for the heteroskedastic variances.

## 6 Conclusion

This paper presents an investigation of online linear experimental design problems with heteroskedastic noise. We propose a two-phase sample splitting procedure for estimating the unknown noise variance parameters based on G-optimal experimental designs and show error bounds that scale efficiently with the dimension of the problem. The proposed approach is then applied to fixed confidence transductive best-arm and level set identification with heteroskedastic noise, where we present instance-dependent lower bounds with provably near-optimal algorithms.

This paper leads to a number of interesting future research directions. For example, there may be situations where the noise structure is not fully characterized by $\Sigma^*$. Future work can explore the development of methods that handle more general noise structures (including correlated or heavy-tailed noise) while remaining near-optimal. Furthermore, it is possible that an integrated approach to variance estimation and adaptive experimentation with covariates would be more sample efficient than our two-step method. Lastly, it would be interesting to extend our algorithm to nonlinear response models.

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

# Contents

# A    Background on Experimental Designs and Rounding

We use optimal experiment design [29, 42] to minimize uncertainty in the estimator of interest. Consider the WLS estimator with the weight matrix $W_T = \Sigma_T^{-1}$ so that $\mathbb{V}(\widehat{\theta}_{\mathrm{WLS}}) = (X_T^\top \Sigma_T^{-1} X_T)^{-1} = (\sum_{t=1}^T \sigma_{x_t}^{-2} x_t x_t^\top)^{-1}$. Let $\lambda_x$ denote the proportion of the total samples $T$ given to the measurement vector $x \in \mathcal{X}$ so that $\mathbb{V}(\widehat{\theta}_{\mathrm{WLS}}) = (\sum_{x \in \mathcal{X}} \lambda_x \sigma_x^{-2} x x^\top)^{-1}/T$. We wish to take $T$ samples so as to construct the design $\sum_{x \in \mathcal{X}} T \lambda_x \sigma_x^{-2} x x^\top$, however $T \lambda_x$ is often not an integer. As has been discussed at depth [46, 12, 35, 42, 3], efficient rounding schemes can solve this problem. Given some constant tolerance threshold $\epsilon$, a design $\lambda \in P_{\mathcal{X}}$, and a minimum sample size $r(\epsilon)$, efficient rounding procedures return a fixed allocation $\{x_t\}_{t=1}^T$ that yields a $(1 + \epsilon)$ approximation to the intended design. A simple, well known procedure with $r(\epsilon) \leq \{d(d+1) + 2\}/\epsilon$ comes from Pukelsheim [42], while the scheme of Allen-Zhu et al. [3] yields $r(\epsilon) \leq \mathcal{O}(d/\epsilon^2)$. In our experimental design algorithms, we leverage the aforementioned rounding schemes.

We adapt the guarantees of [3] for Alg. 1 as an example in the following proposition. For sample size $N$, define $\mathcal{X}^*$ as a set composed of the elements of $\mathcal{X} \subset \mathbb{R}^d$ duplicated $N$ times and $\mathcal{S}_N = \left\{ s \in \{0, 1, \ldots, N\}^{N|\mathcal{X}|} : \sum_{t=1}^{N|\mathcal{X}|} s_i \leq N \right\}$ as the set of discrete sample allocations.

**Proposition A.1.** Suppose $\epsilon \in (0, 1/3)$ and $N \geq 5d/\epsilon^2$. Let $\lambda \in P_{\mathcal{X}}$, then in polynomial-time (in $n$ and $d$), we can round $\lambda N$ to an integral solution $\widehat{\lambda} \in \mathcal{S}_N$ satisfying

$$\max_{x \in \mathcal{X}} x^\top \left( \sum_{x' \in \mathcal{X}^*} \widehat{\lambda}_{x'} x' x'^\top \right)^{-1} x \leq (1 + 6\epsilon) \min_{\lambda \in P_{\mathcal{X}}} \max_{x \in \mathcal{X}} x^\top \left( \sum_{x' \in \mathcal{X}} N\lambda_{x'} x' x'^\top \right)^{-1} x.$$

*Proof.* This follows directly from Theorem 2.1 in [3] letting $\mathcal{C}_N = \left\{ s \in [0, N]^{N|\mathcal{X}|} : \sum_{i=1}^{N|\mathcal{X}|} s_i \leq N \right\}$ be a continuous relaxation of $\mathcal{S}_N$ and setting $k = N$ and $n = N|\mathcal{X}|$. ∎

Alternatively, one could use the robust inverse propensity score (RIPS) estimator [6], which avoids the need for rounding schemes via robust mean estimation.

# B  Technical Preliminaries

For our analysis, we do not assume that the error, $\eta$, in Eq. 1 is Gaussian. We generalize by extending to strictly sub-Gaussian noise, in which the variance $\sigma_x^2$ is equal to the optimal variance proxy for $\eta_t$, $\forall t \in \mathbb{N}$. This paradigm allows for noise distributions such as the symmetrized Beta, symmetrized Gamma, and Uniform.

The following definitions and propositions will be used in the proofs of Appendices D-F.

**Definition B.1.** A real-valued random variable $X$ is *sub-Gaussian* if there exists a positive constant $\sigma^2$ such that
$$\mathbb{E}\left(e^{\lambda X}\right) \leq e^{\lambda^2 \sigma^2 / 2}$$
for all $\lambda \geq 0$.

**Proposition B.1** (Equation 2.9, [49]). Let $X$ be a sub-Gaussian random variable with sub-Gaussian parameter $\sigma^2$ and mean $\mathbb{E}(X) = \mu$. Then, for all $t > 0$, we have
$$\mathbb{P}\left(|X - \mu| \geq t\right) \leq 2e^{-\frac{t^2}{2\sigma^2}}.$$

**Definition B.2.** A real-valued random variable $X$ with mean $\mathbb{E}(X) = \mu$ is *sub-exponential* if there are non-negative parameters $(\nu, \alpha)$ such that
$$\mathbb{E}\left\{e^{\lambda(X-\mu)}\right\} \leq e^{\frac{\nu^2 \lambda^2}{2}}, \ \forall |\lambda| < \frac{1}{\alpha}.$$

**Proposition B.2** (Proposition 2.9, [49]). Suppose that $X$ is sub-exponential with parameters $(\nu, \alpha)$ and mean $\mathbb{E}(X) = \mu$. Then, for $t > 0$,
$$\mathbb{P}\left(X - \mu \geq t\right) \leq \begin{cases} e^{\frac{-t^2}{\nu^2}}, & \text{if } 0 \leq t \leq \frac{\nu^2}{\alpha} \\ e^{-\frac{t}{\alpha}}, & \text{if } t > \frac{\nu^2}{\alpha}. \end{cases}$$

This implies that for $\delta \in (0, 1)$,
$$\mathbb{P}\left[|X - \mu| < \max\left\{\sqrt{2\log(2/\delta)\nu^2}, 2\log(2/\delta)\alpha\right\}\right] = 1 - \delta.$$

**Proposition B.3** (Equation 37 in Appendix B, [19]). Let $X$ be a sub-Gaussian random variable with sub-Gaussian parameter $\sigma^2$. Then $X^2$ is sub-exponential with parameters $(\nu = 4\sigma^2\sqrt{2}, \alpha = 4\sigma^2)$.

**Proposition B.4** (Equation 2.18, [49]). Suppose that $X_i$ for $i \in \{1, 2, \ldots, n\}$ are sub-exponential with parameters $(\nu_i, \alpha_i)$ such that $\mathbb{E}(X_i) = \mu_i$. Then, $\sum_{i=1}^n (X_i - \mu_i)$ is sub-exponential with parameters $(\nu_*, \alpha_*)$ such that
$$\alpha_* = \max_{i=1,2,\ldots,n} \alpha_i \ \text{and} \ \nu_* := \sqrt{\sum_{i=1}^n \nu_i^2}.$$

**Definition B.3.** A *G-optimal design* [28], $\lambda^* \in P_{\mathcal{X}}$, for a set of arms $\mathcal{X} \subset \mathbb{R}^d$ is such that

$$\lambda^* = \arg\min_{\lambda \in P_{\mathcal{X}}} \max_{x \in \mathcal{X}} x^\top \left( \sum_{x \in \mathcal{X}} \lambda_x xx^\top \right)^{-1} x.$$

**Proposition B.5** (Lemma 4.6, [28]). For any finite $\mathcal{X} \subset \mathbb{R}^d$ with $\text{span}(\mathcal{X}) = \mathbb{R}^d$,

$$d = \min_{\lambda \in P_{\mathcal{X}}} \max_{x \in \mathcal{X}} x^\top \left( \sum_{x \in \mathcal{X}} \lambda_x xx^\top \right)^{-1} x.$$

This implies that if we sample each arm $x \in \mathcal{X} \lceil \tau \lambda_x^* \rceil$ times, then

$$\max_{x \in \mathcal{X}} x^\top \left( \sum_{x \in \mathcal{X}} \lceil \tau \lambda_x^* \rceil xx^\top \right)^{-1} x \leq \frac{1}{\tau} \max_{x \in \mathcal{X}} x^\top \left( \sum_{x \in \mathcal{X}} \lambda_x^* xx^\top \right)^{-1} x = \frac{d}{\tau}.$$

**Definition B.4.** The *Forbenius norm* of $A \in \mathbb{R}^{n \times m}$ is

$$\|A\|_F = \sqrt{\text{tr}(A^\top A)}.$$

**Definition B.5.** The *Spectral norm* of $A \in \mathbb{R}^{n \times m}$ is

$$\|A\|_2 = \sup_{x \neq 0} \frac{\|Ax\|_2}{\|x\|_2}.$$

**Proposition B.6** (Matrix Norm Properties). For $A \in \mathbb{R}^{n \times m}$, we have sub-multiplicativity for the Frobenius norm:

$$\|A^\top A\|_F \leq \|A\|_F^2,$$

and a bound on the Spectral norm:

$$\|A\|_2 \leq \|A\|_F.$$

**Proposition B.7** (Corollary 4.7, [61]). Let $\xi_1, \ldots, \xi_n$ be zero-mean $\sigma^2$-sub-Gaussian and $\Delta \in \mathbb{R}^{n \times n}$. For any $t > 0$,

$$P\left[ \xi^\top \Delta \xi \geq \sigma^2 \left\{ \text{tr}(\Delta) + 2\text{tr}(\Delta^2 t)^{1/2} + 2\|\Delta\|_2 t \right\} \right] \leq e^{-t},$$

which implies that with probability $1 - \delta$ for $\delta \in (0, 1)$,

$$\xi^\top \Sigma \xi \leq \sigma^2 \left[ \text{tr}(\Delta) + 2\text{tr}\left\{ \Delta^2 \log(1/\delta) \right\}^{1/2} + 2\|\Delta\|_2 \log(1/\delta) \right].$$

**Proposition B.8** (Equation 20.3, [32]). Assume the setup of Eq. 1 and that a data set of size $\Gamma$, $\mathcal{D} = \{x_t, y_t\}_{t=1}^\Gamma$, has been collected through a fixed design. Define $V_\Gamma = \sum_{t=1}^\Gamma x_t x_t^\top$ and construct the least squares estimator for $\mathcal{D}$, $\widehat{\theta}_\Gamma = V_\Gamma^{-1} \left( \sum_{t=1}^\Gamma x_t y_t \right)$. Then, with probability $1 - \delta$ for $\delta \in (0, 1)$,

$$\|\widehat{\theta}_\Gamma - \theta^*\|_{V_\Gamma} \leq 2\sqrt{2\left\{ d\log(6) + \log(1/\delta) \right\}}.$$

## C  Baseline Variance Estimation Procedures

These are the algorithms we reference in the theoretical and empirical comparisons of Section 3.

**The Uniform Estimator.** The Uniform Estimator draws arms uniformly at random and uses all samples to construct estimators $\widehat{\theta}_\Gamma$ of $\theta^*$ and $\widehat{\Sigma}_\Gamma$ of $\Sigma^*$.

**The Separate Arm Estimator.** Define $\mathcal{U}$ as the set of subsets of $\mathcal{W}$ of size $M = d(d + 1)/2$, and for $U \in \mathcal{U}$ construct $\Phi_U$ such that its rows are composed of $\phi_x \in U$. Let $\zeta_{\min}(A)$ be the minimum singular value of matrix $A \in \mathbb{R}^{d \times d}$. The Seperate Arm Estimator picks a set of $M$ arms such that

$$U^* = \arg\max_{U \in \mathcal{U}} \zeta_{\min}(\Phi_U^{-1}).$$

It then splits $\Gamma$ samples evenly between these arms, estimates the sample variance for each and finds the least squares estimator $\widehat{\Sigma}_\Gamma^{\text{SA}}$.

---

**Algorithm 3:** Uniform Estimator of Heteroskedastic Variance

---

**Result:** Find $\widehat{\Sigma}_\Gamma^{\mathrm{UE}}$

**1 Input:** Arms $\mathcal{X} \subseteq \mathbb{R}^d$ and $\Gamma \in \mathbb{N}$

**2** Sample arms uniformly at random from $\mathcal{X}$

**3** Define $A^* = \sum_{t=1}^\Gamma x_t x_t^\top$ and $b^* = \sum_{t=1}^\Gamma x_t y_t$ and estimate $\widehat{\theta}_\Gamma = A^{*-1} b^*$.

**4** Define $A^\dagger = \sum_{t=1}^\Gamma \phi_{x_t} \phi_{x_t}^\top$ and $b^\dagger = \sum_{t=1}^\Gamma \phi_{x_t} \left( y_t - x_t^\top \widehat{\theta}_\Gamma \right)^2$.

**5 Output:** $\widehat{\Sigma}_\Gamma^{\mathrm{UE}} = A^{\dagger-1} b^\dagger$.

---

---

**Algorithm 4:** Seperate Arm Estimator of Heteroskedastic Variance

---

**Result:** Find $\widehat{\Sigma}_\Gamma^{\mathrm{SA}}$

**1 Input:** Arms $\mathcal{X} \subseteq \mathbb{R}^d$ and $\Gamma \in \mathbb{N}$

**2** Define $U^* = \arg\max_{U \in \mathcal{U}} \zeta_{\min}(\Phi_U^{-1})$.

**3** Equally distribute $\Gamma$ samples among arms $\mathcal{X}_{U^*} = \{m : \phi_m \in U^*\}$ and observe rewards $\{y_{m,t}\}_{t=1}^{\Gamma/M}$ for $m \in \mathcal{X}_{U^*}$

**4** Define the sample average for each arm $\bar{y}_m = (M/\Gamma) \sum_{t=1}^{\Gamma/M} y_{m,t}$

**5 Output:** $\widehat{\Sigma}_\Gamma^{SA} \leftarrow \arg\min_{\boldsymbol{s} \in \mathbb{R}^M} \sum_{m \in \mathcal{X}_{U^*}} \sum_{t=1}^{\Gamma/M} \left\{ \phi_m^\top \boldsymbol{s} - (\bar{y}_m - y_{m,t})^2 \right\}^2$

---

# D  Proofs of Variance Estimation

In this appendix, we analyze the maximum absolute error of estimates associated with the Seperate Arm Estimator and the HEAD Estimator. Recall that $M := d(d+1)/2$. Define $\mathcal{W} = \{\phi_x, \forall x \in \mathcal{X}\} \subset \mathbb{R}^M$ where $\phi_x = \mathrm{vec}(xx^\top) G$ and $G \in \mathbb{R}^{d^2 \times M}$ is the duplication matrix such that $G\mathrm{vech}(xx^\top) = \mathrm{vec}(xx^\top)$ with $\mathrm{vech}(\cdot)$ denoting the half-vectorization.

## D.1  Analysis of the Separate Arm Estimator

We begin by proving a general result bounding the absolute error of the sample variance for any sub-Gaussian random variable with parameter $\gamma^2$.

**Proposition D.1.** Let $X$ be a $\gamma^2$-sub-Gaussian random variable with mean $\mu$. Assume that we have collected $n$ independent copies of $X$, $\{X_1, X_2, \ldots, X_n\}$, and label the sample mean as $\bar{X} = \sum_{i=1}^n X_i/n$. Defining the sample variance as $\widehat{\sigma}^2 = \sum_{i=1}^n (X_i - \bar{X})^2/n$, the true variance as $\sigma^2 = \mathbb{E}(X_i^2) - \mu^2$ and $\delta \in (0,1)$, we find that with probability greater than $1 - \delta$,

$$|\sigma^2 - \widehat{\sigma}^2| \leq \frac{5}{2} \max\left\{ \frac{4\sqrt{2}\gamma^2 \log(4/\delta)}{n}, \sqrt{\frac{32\gamma^4 \log(4/\delta)}{n}} \right\}.$$

*Proof.* We begin by upper bounding the absolute error using the triangle inequality,

$$
\begin{aligned}
\left| \sigma^2 - \widehat{\sigma}^2 \right| &= \left| \sigma^2 - \frac{\sum_{i=1}^n (X_i - \bar{X})^2}{n} \right| \\
&= \left| \mathbb{E}(X_i^2) - \mu^2 - \frac{\sum_{i=1}^n X_i^2 - 2X_i \bar{X} + \bar{X}^2}{n} \right| \\
&= \left| \mathbb{E}(X_i^2) - \mu^2 - \frac{\sum_{i=1}^n X_i^2}{n} + 2\bar{X}^2 - \bar{X}^2 \right| \\
&= \left| \mathbb{E}(X_i^2) - \frac{\sum_{i=1}^n X_i^2}{n} + \bar{X}^2 - \frac{\sigma^2}{n} - \mu^2 + \frac{\sigma^2}{n} \right| \\
&\leq \underbrace{\left| \mathbb{E}(X_i^2) - \frac{\sum_{i=1}^n X_i^2}{n} \right|}_{\mathbf{A}} + \underbrace{\left| \bar{X}^2 - \frac{\sigma^2}{n} - \mu^2 \right|}_{\mathbf{B}} + \underbrace{\left| \frac{\sigma^2}{n} \right|}_{\mathbf{C}}.
\end{aligned}
$$

We first bound quantity $\mathbf{A}$. Leveraging Proposition B.3, we establish that $X_i^2 - \mathbb{E}(X_i^2)$ is sub-exponential with parameters $(4\gamma^2\sqrt{2}, 4\gamma^2)$. We can then use Proposition B.4 to analyze the sum of independent sub-exponential random variables, finding that $\sum_{i=1}^n \left\{ X_i^2 - \mathbb{E}(X_i^2) \right\} / n$ is sub-exponential with parameters $(4\gamma^2\sqrt{2}/\sqrt{n}, 4\gamma^2/n)$. Finally, we invoke Proposition B.2 to bound quantity $\mathbf{A}$ with probability $1 - \delta/2$,

$$
\left| \mathbb{E}(X_i^2) - \frac{\sum_{i=1}^n X_i^2}{n} \right| \leq \max \left\{ \frac{4\gamma^2 \log(4/\delta)}{n}, \sqrt{\frac{32\gamma^4 \log(4/\delta)}{n}} \right\}.
$$

Now we bound quantity $\mathbf{B}$. Appealing to Proposition B.3, we find that $\bar{X}^2 - \frac{\sigma^2}{n} - \mu^2$ is sub-exponential with parameters $(4\sqrt{2}\gamma^2/n, 4\gamma^2/n)$. We then use Proposition B.2 to bound $\mathbf{B}$ with probability $1 - \delta/2$,

$$
\begin{aligned}
\left| \bar{X}^2 - \frac{\sigma^2}{n} - \mu^2 \right| &\leq \max \left\{ \frac{4\gamma^2 \log(4/\delta)}{n}, \sqrt{\frac{32\gamma^4 \log(4/\delta)}{n^2}} \right\} \\
&= \max \left\{ \frac{4\gamma^2 \log(4/\delta)}{n}, \frac{4\gamma^2 \log(4/\delta)}{n} \frac{\sqrt{2}}{\sqrt{\log(4/\delta)}} \right\} \\
&\overset{(a)}{\leq} \frac{4\sqrt{2}\gamma^2 \log(4/\delta)}{n},
\end{aligned}
$$

where $(a)$ follows because $\sqrt{2}/\sqrt{\log(4/\delta)} \leq \sqrt{2}$. Finally, we combine quantities $\mathbf{A}, \mathbf{B}$ and $\mathbf{C}$, apply a union bound, and conclude that with probability $1 - \delta$ for $\delta \in (0, 1)$,

$$
\begin{aligned}
\left| \sigma^2 - \widehat{\sigma}^2 \right| &\leq \left| \mathbb{E}(X_i^2) - \frac{\sum_{i=1}^n X_i^2}{n} \right| + \left| \bar{X}^2 - \frac{\sigma^2}{n} - \mu^2 \right| + \left| \frac{\sigma^2}{n} \right| \\
&\leq \max \left\{ \frac{4\gamma^2 \log(4/\delta)}{n}, \sqrt{\frac{32\gamma^4 \log(4/\delta)}{n}} \right\} + \frac{4\sqrt{2}\gamma^2 \log(4/\delta)}{n} + \frac{\gamma^2}{n} \\
&\leq \frac{5}{2} \max \left\{ \frac{4\sqrt{2}\gamma^2 \log(4/\delta)}{n}, \sqrt{\frac{32\gamma^4 \log(4/\delta)}{n}} \right\}.
\end{aligned}
$$

∎

We can now use Proposition D.1 to control the sample variance estimates of the arms chosen by the Separate Arm Estimator. Theorem D.1 derives a concentration bound of the maximum absolute error of $\widetilde{\sigma}_x^2 = \min \left\{ \max \left\{ \sigma_{\min}^2, x^\top \widehat{\Sigma}_\Gamma^{\mathrm{SA}} x \right\}, \sigma_{\max}^2 \right\}$. This bound, which is $\mathcal{O}(d^4)$, scales unfavorably with dimension as highlighted in the theoretical and empirical comparisons conducted in Section 3.

**Theorem D.1.** Set $\widehat{\Sigma}_{\Gamma}^{\mathrm{SA}}$ as the output of Alg. 4 and define $B$ such that for all $x \in \mathcal{X}$, $\|x\|_2 \leq B$. Defining $\widetilde{\sigma}_x^2 = \min\left\{\max\left\{\sigma_{\min}^2, x^\top \widehat{\Sigma}_{\Gamma}^{\mathrm{SA}} x\right\}, \sigma_{\max}^2\right\}$, $\sigma_x^2 = x^\top \Sigma^* x$ and $\delta \in [0,1]$, for any $x \in \mathcal{X}$,

$$\mathbb{P}\left[|\widetilde{\sigma}_x^2 - \sigma_x^2| \leq \frac{5B^2\sqrt{2}}{\zeta_{\min}(\Phi_{U^*})}\max\left\{\frac{4\sqrt{2}\sigma_{\max}^2 d^3 \log(8M/\delta)}{\Gamma}, \sqrt{\frac{32\sigma_{\max}^4 d^4 \log(8M/\delta)}{\Gamma}}\right\}\right] \geq 1 - \delta/2.$$

Note that we set the probability bound to $1 - \delta/2$ instead of $1 - \delta$ because we want to be able to leverage these variance estimation bounds with the identification algorithms of Section 4 (and so are anticipating another union bound).

*Proof.* Label $M = d(d+1)/2$ and $\mathcal{X}_{U^*} = \{m : \phi_m \in U^*\}$, where $U^*$ is defined in Alg. 4. Additionally, in Alg. 4 we observe $\{y_{m,t}\}_{t=1}^{\Gamma/M}$ for each arm $m \in \mathcal{X}_{U^*}$. Define the sample average for $m \in \mathcal{X}_{U^*}$ as

$$\bar{y}_m = \frac{\sum_{t=1}^{\Gamma/M} y_{m,t}}{\Gamma/M}.$$

We obtain an estimate of $\Sigma^*$ via

$$\mathrm{vech}(\widehat{\Sigma}_{\Gamma}^{\mathrm{SA}}) = \arg\min_{\boldsymbol{s} \in \mathbb{R}^M} \sum_{m \in \mathcal{X}_{U^*}} \sum_{t=1}^{\Gamma/M} \left\{\langle \phi_m, \boldsymbol{s}\rangle - (\bar{y}_m - y_{m,t})^2\right\}^2. \tag{5}$$

Label $z_{m,t} = (\bar{y}_m - y_{m,t})^2 \in \mathbb{R}$, $\Psi^* = \mathrm{vech}(\Sigma^*) \in \mathbb{R}^M$, $\widetilde{\Psi}_{\Gamma} = \mathrm{vech}(\widehat{\Sigma}_{\Gamma}^{\mathrm{SA}}) \in \mathbb{R}^M$ and

$$\bar{z}_m = \frac{\sum_{t=1}^{\Gamma/M} z_{m,t}}{\Gamma/M}.$$

We reformulate Eq. 5 as follows

$$\widetilde{\Psi}_{\Gamma} = \arg\min_{\boldsymbol{s} \in \mathbb{R}^M} \sum_{m \in \mathcal{X}_{U^*}} \sum_{t=1}^{\Gamma/M} \left(\langle \phi_m, \mathbf{s}\rangle - z_{m,t}\right)^2$$

$$= \arg\min_{\boldsymbol{s} \in \mathbb{R}^M} \sum_{m \in \mathcal{X}_{U^*}} \sum_{t=1}^{\Gamma/M} \langle \phi_m, \mathbf{s}\rangle^2 - 2z_{m,t}\langle \phi_m, \mathbf{s}\rangle + z_{m,t}^2$$

$$= \arg\min_{\boldsymbol{s} \in \mathbb{R}^M} \sum_{m \in \mathcal{X}_{U^*}} (\Gamma/M)\langle \phi_m, \mathbf{s}\rangle^2 - 2\langle \phi_m, \mathbf{s}\rangle \sum_{t=1}^{\Gamma/M} z_{m,t} + \sum_{t=1}^{\Gamma/M} z_{m,t}^2$$

$$= \arg\min_{\boldsymbol{s} \in \mathbb{R}^M} \sum_{m \in \mathcal{X}_{U^*}} \langle \phi_m, \mathbf{s}\rangle^2 - 2\langle \phi_m, \mathbf{s}\rangle \bar{z}_m + \frac{\sum_{t=1}^{\Gamma/M} z_{m,t}^2}{\Gamma/M}$$

$$\overset{(a)}{=} \arg\min_{\boldsymbol{s} \in \mathbb{R}^M} \sum_{m \in \mathcal{X}_{U^*}} \langle \phi_m, \mathbf{s}\rangle^2 - 2\langle \phi_m, \mathbf{s}\rangle \bar{z}_m + \bar{z}_m^2$$

$$= \arg\min_{\boldsymbol{s} \in \mathbb{R}^M} \sum_{m \in \mathcal{X}_{U^*}} \left(\langle \phi_m, \mathbf{s}\rangle - \bar{z}_m\right)^2,$$

where $(a)$ follows because $\sum_{t=1}^{\Gamma/M} z_{m,t}^2/(\Gamma/M)$ and $\bar{z}_m^2$ do not depend on $\mathbf{s}$. Defining $\bar{z} = [\bar{z}_m]_{m \in \mathcal{X}_{U^*}} \in \mathbb{R}^M$, Eq. 5 becomes

$$\widetilde{\Psi}_{\Gamma} = \arg\min_{\boldsymbol{s} \in \mathbb{R}^M} \sum_{m \in \mathcal{X}_{U^*}} \left(\langle \phi_m, \mathbf{s}\rangle - \bar{z}_m\right)^2 = \arg\min_{\boldsymbol{s} \in \mathbb{R}^M} \|\Phi_{U^*}\mathbf{s} - \bar{z}\|_2^2.$$

We find that

$$\|\Phi_{U^*}(\widetilde{\Psi}_{\Gamma} - \Psi^*)\|_2 = \|\Phi_{U^*}\widetilde{\Psi}_{\Gamma} - \Phi_{U^*}\Psi^* - \bar{z} + \bar{z}\|_2$$

$$= \|\Phi_{U^*}\widetilde{\Psi}_{\Gamma} - \bar{z} + \bar{z} - \Phi_{U^*}\Psi^*\|_2$$

$$\leq \|\Phi_{U^*}\widetilde{\Psi}_{\Gamma} - \bar{z}\|_2 + \|\bar{z} - \Phi_{U^*}\Psi^*\|_2$$

$$\leq \|\Phi_{U^*}\Psi^* - \bar{z}\|_2 + \|\bar{z} - \Phi_{U^*}\Psi^*\|_2$$

$$= 2\|\Phi_{U^*}\Psi^* - \bar{z}\|_2.$$

Consequently, we can focus on bounding $\|\Phi_{U^*}\Psi^* - \bar{z}\|_2$ in order to control $\|\Phi_{U^*}(\widetilde{\Psi}_\Gamma - \Psi^*)\|_2$. Note that the $m^{th}$ entry of $(\Phi_{U^*}\Psi^* - \bar{z})$ is $(m^\top \Sigma^* m - \bar{z}_m)$ and that $\bar{z}_m$ is the sample variance of observations $\{y_{m,t}\}_{t=1}^{\Gamma/M}$. Using Proposition D.1 and the fact that the $y_{m,t}$ are $\sigma_{\max}^2$-sub-Gaussian, we show that with probability $1 - \delta/(2M)$,

$$|\phi_m^\top \Psi^* - \bar{z}_m| \le \frac{5}{2}\max\left\{\frac{4\sqrt{2}\sigma_{\max}^2 M \log(8M/\delta)}{\Gamma}, \sqrt{\frac{32\sigma_{\max}^4 M \log(8M/\delta)}{\Gamma}}\right\}. \tag{6}$$

We apply Eq. 6 to quantity $\|\Phi_{U^*}\Psi^* - \bar{z}\|_2$ and a union bound to find that with probability $1 - \delta/2$,

$$\|\Phi_{U^*}\Psi^* - \bar{z}\|_2 \le \sqrt{M}\frac{5}{2}\max\left\{\frac{4\sqrt{2}\sigma_{\max}^2 M \log(8M/\delta)}{\Gamma}, \sqrt{\frac{32\sigma_{\max}^4 M \log(8M/\delta)}{\Gamma}}\right\}$$

$$= \frac{5}{2}\max\left\{\frac{4\sqrt{2}\sigma_{\max}^2 M\sqrt{M} \log(8M/\delta)}{\Gamma}, \sqrt{\frac{32\sigma_{\max}^4 M^2 \log(8M/\delta)}{\Gamma}}\right\}.$$

Consequently,

$$\|\Phi_{U^*}(\widetilde{\Psi}_\Gamma - \Psi^*)\|_2 \le 5\max\left\{\frac{4\sqrt{2}\sigma_{\max}^2 M\sqrt{M} \log(8M/\delta)}{\Gamma}, \sqrt{\frac{32\sigma_{\max}^4 M^2 \log(8M/\delta)}{\Gamma}}\right\}.$$

We now analyze the absolute error of the estimate $\widetilde{\sigma}_x$ for an arbitrary action $x \in \mathcal{X}$. Since $\Phi_{U^*}$ is non-singular by construction, we know that for $q := (\Phi_{U^*}^{-1})^\top \phi_x \in \mathbb{R}^M$,

$$
\begin{aligned}
|\widetilde{\Psi}_\Gamma^\top \phi_x - \sigma_x^2| &= \|(\widetilde{\Psi}_\Gamma - \Psi^*)^\top \phi_x\|_2 \\
&= \|(\widetilde{\Psi}_\Gamma - \Psi^*)^\top \Phi_{U^*}^\top q\|_2 \\
&\le \|(\widetilde{\Psi}_\Gamma - \Psi^*)^\top \Phi_{U^*}^\top\|_2 \|q\|_2 \\
&= \|\Phi_{U^*}(\widetilde{\Psi}_\Gamma - \Psi^*)\|_2 \|(\Phi_{U^*}^{-1})^\top \phi_x\|_2 \\
&\le \|\Phi_{U^*}(\widetilde{\Psi}_\Gamma - \Psi^*)\|_2 \|(\Phi_{U^*}^{-1})^\top\|_2 \|\phi_x\|_2 \\
&\overset{(a)}{=} \frac{2}{\zeta_{\min}(\Phi_{U^*})}\|\Phi_{U^*}(\widetilde{\Psi}_\Gamma - \Psi^*)\|_2 \|\phi_x\|_2,
\end{aligned}
$$

where $(a)$ follows from the definition of the matrix norm. We now leverage the nature of the duplication matrix, $G$, and the fact that $\|x\|_2 \le B$ to bound $\|\phi_x\|_2^2$.

$$
\begin{aligned}
\|\phi_x\|_2^2 &= \text{vec}(xx^\top)^\top GG^\top \text{vec}(xx^\top) \\
&= \text{vec}(xx^\top)^\top G(G^\top G)(G^\top G)^{-1}G^\top \text{vec}(xx^\top) \\
&= \text{vec}(xx^\top)^\top G(G^\top G)^{-1/2}(G^\top G)(G^\top G)^{-1/2}G^\top \text{vec}(xx^\top) \\
&\overset{(a)}{\le} 2\text{vec}(xx^\top)^\top G(G^\top G)^{-1}G^\top \text{vec}(xx^\top) \\
&\overset{(b)}{=} 2\text{vec}(xx^\top)^\top \text{vec}(xx^\top) \\
&= 2x^\top xx^\top x \\
&\le 2B^4,
\end{aligned}
$$

where $(a)$ follows from the fact that $G^\top G \preceq 2I$ as shown in Equation (61) of [40], and $(b)$ uses the fact that $G(G^\top G)^{-1}G^\top \text{vec}(xx^\top) = \text{vec}(xx^\top)$ as shown in Equations (54) and (52) of [40]. Consequently, with probability greater than $1 - \delta/2$, we have that for any $x \in \mathcal{X}$,

$$
\begin{aligned}
|\widetilde{\sigma}_x^2 - \sigma_x^2| &\le \frac{5B^2\sqrt{2}}{\zeta_{\min}(\Phi_{U^*})}\max\left\{\frac{4\sqrt{2}\sigma_{\max}^2 M\sqrt{M} \log(8M/\delta)}{\Gamma}, \sqrt{\frac{32\sigma_{\max}^4 M^2 \log(8M/\delta)}{\Gamma}}\right\} \\
&\le \frac{5B^2\sqrt{2}}{\zeta_{\min}(\Phi_{U^*})}\max\left\{\frac{4\sqrt{2}\sigma_{\max}^2 d^3 \log(8M/\delta)}{\Gamma}, \sqrt{\frac{32\sigma_{\max}^4 d^4 \log(8M/\delta)}{\Gamma}}\right\}.
\end{aligned}
$$

∎

## D.2 Analysis of the `HEAD` Estimator

We now present the maximum absolute error guarantees of the `HEAD` procedure, Alg. 1.

*Proof of Theorem 3.1.* In this proof, we bound the estimation error $|\widehat{\sigma}_x^2 - \sigma_x^2|$, $\forall x \in \mathcal{X}$. Define $\sigma_{\max}^2 = \max_{x \in \mathcal{X}} \sigma_x^2$, $\Psi^* = \text{vech}(\Sigma^*)$, $N_1 = \{1, \ldots, \Gamma/2\}$, $N_2 = \{\Gamma/2 + 1, \ldots, \Gamma\}$ and

$$\widehat{\Psi}_\Gamma = \arg \min_{\boldsymbol{s} \in \mathbb{R}^M} \sum_{t \in N_2} \left\{ \boldsymbol{s}^\top \phi_{x_t} - (y_t - x_t^\top \widehat{\theta}_{\Gamma/2})^2 \right\}^2,$$

where $\widehat{\Psi}_\Gamma = \text{vech}(\widehat{\Sigma}_\Gamma)$. Furthermore, define vector $e$ with entries $e_t = (y_t - x_t^\top \widehat{\theta}_{\Gamma/2})^2$ and $\Phi$ such that the rows of $\Phi$ are composed of $\{\phi_{x_t}\}_{t \in N_2}$. We find that

$$\widehat{\Psi}_\Gamma = (\Phi^\top \Phi)^{-1} \Phi^\top e.$$

For any $\phi_x \in \mathcal{W}$, we are interested in bounding $\left| \phi_x^\top (\widehat{\Psi}_\Gamma - \Psi^*) \right|$,

$$|\phi_x^\top (\widehat{\Psi}_\Gamma - \Psi^*)| = \left| \phi_x^\top \left\{ (\Phi^\top \Phi)^{-1} \Phi^\top e - \Psi^* \right\} \right|$$
$$= | \underbrace{\phi_x^\top (\Phi^\top \Phi)^{-1} \Phi^\top}_{z^\top} (e - \Phi \Psi^*)|.$$

Define $z^\top = \phi_x^\top (\Phi^\top \Phi)^{-1} \Phi^\top$ and $X$ such that the rows of $X$ are composed of $\{x_t\}_{t \in N_2}$. We can bound this expression with the triangle inequality as

$$|\phi_x^\top (\widehat{\Psi}_\Gamma - \Psi^*)| = \left| \sum_{t \in N_2} z_t \left\{ x_t^\top (\widehat{\theta}_{\Gamma/2} - \theta^*) \right\}^2 + \sum_{t \in N_2} z_t (\eta_t^2 - \phi_t^\top \Psi^*) + \sum_{t \in N_2} 2 z_t x_t^\top (\widehat{\theta}_{\Gamma/2} - \theta^*) \eta_t \right|$$

$$\leq \left| \sum_{t \in N_2} z_t \left\{ x_t^\top (\widehat{\theta}_{\Gamma/2} - \theta^*) \right\}^2 \right| + \left| \sum_{t \in N_2} z_t (\eta_t^2 - \phi_t^\top \Psi^*) \right| + \left| \sum_{t \in N_2} 2 z_t x_t^\top (\widehat{\theta}_{\Gamma/2} - \theta^*) \eta_t \right|.$$

We can simplify the first term as,

$$\left| \sum_{t \in N_2} z_t \left\{ x_t^\top (\widehat{\theta}_{\Gamma/2} - \theta^*) \right\}^2 \right| = \left| \sum_{t \in N_2} z_t \phi_{x_t}^\top \text{vech} \left\{ (\widehat{\theta}_{\Gamma/2} - \theta^*)(\widehat{\theta}_{\Gamma/2} - \theta^*)^\top \right\} \right|$$
$$= \left| \phi_x^\top (\Phi^\top \Phi)^{-1} \Phi^\top \Phi \times \text{vech} \left\{ (\widehat{\theta}_{\Gamma/2} - \theta^*)(\widehat{\theta}_{\Gamma/2} - \theta^*)^\top \right\} \right|$$
$$= \left| x^\top (\widehat{\theta}_{\Gamma/2} - \theta^*)(\widehat{\theta}_{\Gamma/2} - \theta^*)^\top x \right|$$
$$= \left\{ x^\top (\widehat{\theta}_{\Gamma/2} - \theta^*) \right\}^2.$$

Additionally, the third term is equivalent to $2|(\eta \circ z)^\top X (\widehat{\theta}_{\Gamma/2} - \theta^*)|$, where "$\circ$" is element-wise multiplication. Consequently, the absolute error is equal to

$$|\phi_x^\top (\widehat{\Psi}_\Gamma - \Psi^*)| = \underbrace{\left\{ x^\top (\widehat{\theta}_{\Gamma/2} - \theta^*) \right\}^2}_{\mathbf{A}} + \underbrace{\left| \sum_{t \in N_2} z_t (\eta_t^2 - \phi_t^\top \Psi^*) \right|}_{\mathbf{B}} + \underbrace{2 \left| (\eta \circ z)^\top X (\widehat{\theta}_{\Gamma/2} - \theta^*) \right|}_{\mathbf{C}}.$$

Note that to bound $|\phi_x^\top (\widehat{\Psi}_\Gamma - \Psi^*)|$ with probability $1 - \delta/2$, it is necessary to bound quantities $\mathbf{A}$, $\mathbf{B}$ and $\mathbf{C}$ with probability $1 - \delta/6$ in anticipation of a union bound.

**Quantity A.** We sample $\{x_t, t \in N_1\}$ according to the rounded G-optimal design over $\mathcal{X}$ constructed in Stage 1 of Alg. 1. We use the concentration bounds of Proposition B.1, to state that with probability $1 - \delta/6$,

$$\mathbf{A} \leq 2 x^\top \underbrace{\left( \sum_{t \in N_1} x_t x_t^\top \right)^{-1}}_{\mathbf{G}} x \, \sigma_{\max}^2 \log(12 |\mathcal{X}|/\delta).$$

We use Proposition B.5 in combination with the rounding procedure guarantees of Proposition A.1 to bound quantity $\mathbf{G}$. For $\epsilon \in (0, 1/3]$, with probability $1 - \delta/6$,

$$\mathbf{A} \leq \frac{4d(1 + 6\epsilon)\sigma_{\max}^2 \log(12|\mathcal{X}|/\delta)}{\Gamma}.$$

**Quantity B.** According to Proposition B.3, $\eta_t^2$ is sub-exponential with parameters $(4\sqrt{2}\sigma_{\max}^2, 4\sigma_{\max}^2)$. Consequently, by Proposition B.4, we know that $\mathbf{B}$ is sub-exponential with parameters

$$\left( \sqrt{\sum_{t \in N_2} z_t^2 32\sigma_{\max}^4}, \max_{t \in N_2} |z_t| 4\sigma_{\max}^2 \right).$$

We now invoke the G-optimal design over $\mathcal{W}$ space conducted in Stage 2 of Alg. 1 and Propositions B.5 and A.1 to find that for $\epsilon \in (0, 1/3]$,

$$\sum_{t \in N_2} z_t^2 = \sum_{t \in N_2} \phi_x^\top (\Phi^\top \Phi)^{-1} \phi_t \phi_t^\top (\Phi^\top \Phi)^{-1} \phi_x$$
$$= \phi_x^\top (\Phi^\top \Phi)^{-1} \Phi^\top \Phi (\Phi^\top \Phi)^{-1} \phi_x$$
$$\leq \phi_x^\top (\Phi^\top \Phi)^{-1} \phi_x \leq \frac{2d^2(1 + 6\epsilon)}{\Gamma}.$$

Consequently,

$$\sum_{t \in N_2} z_t^2 \leq \frac{2d^2(1 + 6\epsilon)}{\Gamma}, \tag{7}$$

and

$$|z_t| = |\phi_x^\top (\Phi^\top \Phi)^{-1} \phi_t|$$
$$\leq \sqrt{\phi_x^\top (\Phi^\top \Phi)^{-1} \phi_x} \sqrt{\phi_t^\top (\Phi^\top \Phi)^{-1} \phi_t}$$
$$\leq \frac{2d^2(1 + 6\epsilon)}{\Gamma}.$$

Therefore, $\mathbf{B}$ has sub-exponential parameters $(\sqrt{64\sigma_{\max}^4 d^2(1 + 6\epsilon)/\Gamma}, 8\sigma_{\max}^2 d^2(1 + 6\epsilon)/\Gamma)$. Using Proposition B.2, with probability $1 - \delta/6$,

$$\mathbf{B} \leq \max \left\{ \sqrt{\frac{128 \log(12/\delta)\sigma_{\max}^4 d^2(1 + 6\epsilon)}{\Gamma}}, \frac{16 \log(12/\delta)\sigma_{\max}^2 d^2(1 + 6\epsilon)}{\Gamma} \right\}$$
$$= \max \left\{ \sqrt{8}\sigma_{\max} \sqrt{\frac{16 \log(12/\delta)\sigma_{\max}^2 d^2(1 + 6\epsilon)}{\Gamma}}, \frac{16 \log(12/\delta)\sigma_{\max}^2 d^2(1 + 6\epsilon)}{\Gamma} \right\}.$$

**Quantity C.** Construct $V \in \mathbb{R}^{\Gamma/2 \times d}$ such that the rows of $V$ correspond to $\{x_t : t \in N_1\}$. We can upper bound quantity $\mathbf{C}$ using the Cauchy-Schwartz inequality in the following manner,

$$|(\eta \circ z)^\top X(\widehat{\theta}_{\Gamma/2} - \theta^*)| = |(\eta \circ z)^\top X(V^\top V)^{-1/2}(V^\top V)^{1/2}(\widehat{\theta}_{\Gamma/2} - \theta^*)|$$
$$\leq \|(\eta \circ z)^\top X(V^\top V)^{-1/2}\|_2 \|(V^\top V)^{1/2}(\widehat{\theta}_{\Gamma/2} - \theta^*)\|_2.$$

By Proposition B.8, with probability $1 - \delta/12$,

$$\|(V^\top V)^{1/2}(\widehat{\theta}_{\Gamma/2} - \theta^*)\|_2 \leq 2\sqrt{2\sigma_{\max}^2 \{d \log(6) + \log(12/\delta)\}}.$$

Now construct $D_z = \text{diag}(\{z_t : t \in N_2\})$ and express,

$$\|(\eta \circ z)^\top X(V^\top V)^{-1/2}\|_2 = \sqrt{\eta^\top D_z X(V^\top V)^{-1} X^\top D_z \eta}.$$

We first establish that by the G-optimal designs in both $\mathcal{X}$ and $\mathcal{W}$ space along with Propositions B.5 and A.1,

$$
\begin{aligned}
\operatorname{tr}\left\{D_z X (V^\top V)^{-1} X^\top D_z\right\} &= \operatorname{tr}\left\{D_z^2 X (V^\top V)^{-1} X^\top\right\} \\
&= \sum_{t \in N_2} z_t^2 x_t^\top (V^\top V)^{-1} x_t \\
&\leq \frac{2d(1+6\epsilon)}{\Gamma} \sum_{t \in N_2} z_t^2 \\
&\overset{(a)}{\leq} \frac{4d^3(1+6\epsilon)^2}{\Gamma^2},
\end{aligned}
$$

where $\epsilon \in (0, 1/3]$ and $(a)$ follows from Eq. 7. Consequently, we find that

$$
\operatorname{tr}\left\{D_z X (V^\top V)^{-1} X^\top D_z\right\} \leq \frac{4d^3(1+6\epsilon)^2}{\Gamma^2}. \tag{8}
$$

This inequality will be useful in the following derivation. Labeling $\Delta = D_z X (V^\top V)^{-1} X^\top D_z$, we use Proposition B.7 to bound $\eta^\top \Delta \eta$ with probability $1 - \delta/12$ and explain below.

$$
\eta^\top \Delta \eta < \sigma_{\max}^2 \left\{ \operatorname{tr}(\Delta) + 2\|\Delta\|_F \log(12/\delta)^{1/2} + 2\|\Delta\|_2 \log(12/\delta) \right\} \tag{9}
$$

$$
\leq \sigma_{\max}^2 \left\{ \frac{4d^3(1+6\epsilon)^2}{\Gamma^2} + 2\|\Delta^{1/2}\|_F^2 \log(12/\delta)^{1/2} + 2\|\Delta\|_2 \log(12/\delta) \right\} \tag{10}
$$

$$
\leq \sigma_{\max}^2 \left\{ \frac{4d^3(1+6\epsilon)^2}{\Gamma^2} + 2\|\Delta^{1/2}\|_F^2 \log(12/\delta)^{1/2} + 2\|\Delta\|_F \log(12/\delta) \right\} \tag{11}
$$

$$
\leq \sigma_{\max}^2 \left\{ \frac{4d^3(1+6\epsilon)^2}{\Gamma^2} + 2\|\Delta^{1/2}\|_F^2 \log(12/\delta)^{1/2} + 2\|\Delta^{1/2}\|_F^2 \log(12/\delta) \right\} \tag{12}
$$

$$
\leq \sigma_{\max}^2 \left\{ \frac{4d^3(1+6\epsilon)^2}{\Gamma^2} + 2\operatorname{tr}(\Delta) \log(12/\delta)^{1/2} + 2\operatorname{tr}(\Delta) \log(12/\delta) \right\} \tag{13}
$$

$$
\leq \sigma_{\max}^2 \left\{ \frac{4d^3(1+6\epsilon)^2}{\Gamma^2} + 2\frac{4d^3(1+6\epsilon)^2}{\Gamma^2} \log(12/\delta)^{1/2} + 2\frac{4d^3(1+6\epsilon)^2}{\Gamma^2} \log(12/\delta) \right\} \tag{14}
$$

$$
\leq 5\sigma_{\max}^2 \frac{4d^3(1+6\epsilon)^2}{\Gamma^2} \log(12/\delta), \tag{15}
$$

where line (9) follows from Eq. 8, Proposition B.6 and Proposition B.7. In lines (10) and (11), we use Proposition B.6 again and then the definition of the Frobenius norm in line (12). Consequently, with probability $1 - \delta/6$,

$$
\begin{aligned}
\mathbf{C} &= |(\eta \circ z)^\top X (\widehat{\theta}_{\Gamma/2} - \theta^*)| \\
&= |(\eta \circ z)^\top X (V^\top V)^{-1/2} (V^\top V)^{1/2} (\widehat{\theta}_{\Gamma/2} - \theta^*)| \\
&\leq \|(\eta \circ z)^\top X (V^\top V)^{-1/2}\|_2 \|(V^\top V)^{1/2} (\widehat{\theta}_{\Gamma/2} - \theta^*)\|_2 \\
&\leq 2\sqrt{10\sigma_{\max}^4 \frac{4d^3(1+6\epsilon)^2}{\Gamma^2} \log(12/\delta) \left\{d\log(6) + \log(12/\delta)\right\}}.
\end{aligned}
$$

Combining quantities $\mathbf{A}, \mathbf{B}$ and $\mathbf{C}$ together and applying a union bound, we have with probability $1 - \delta/2$,

$$
\begin{aligned}
|\phi_x^\top(\widehat{\Psi}_\Gamma - \Psi^*)| &\leq \max\left\{\sqrt{8}\sigma_{\max}\sqrt{\frac{16\log(12/\delta)\sigma_{\max}^2 d^2(1+6\epsilon)}{\Gamma}}, \frac{16\log(12/\delta)\sigma_{\max}^2 d^2(1+6\epsilon)}{\Gamma}\right\} \\
&\quad + \frac{4d\sigma_{\max}^2\log(12|\mathcal{X}|/\delta)(1+6\epsilon)}{\Gamma} + \\
&\quad + 2\sqrt{10\sigma_{\max}^4\frac{4d^3(1+6\epsilon)^2}{\Gamma^2}\log(12/\delta)\{d\log(6) + \log(12/\delta)\}} \\
&\leq \underbrace{(3/10)\max\left\{\sqrt{8}\sigma_{\max}\sqrt{\frac{16\log(12/\delta)\sigma_{\max}^2 d^2(1+6\epsilon)}{\Gamma}}, \frac{16\log(12/\delta)\sigma_{\max}^2 d^2(1+6\epsilon)}{\Gamma}\right\}}_{\mathbf{W}} \\
&\quad + \underbrace{\frac{4d\sigma_{\max}^2\log(12|\mathcal{X}|/\delta)(1+6\epsilon)}{\Gamma}}_{\mathbf{V}}.
\end{aligned}
$$

We can exploit the dependencies of $\mathbf{V}$ and $\mathbf{W}$ to simplify this bound in two ways under different conditions. Note that we only have dependency on $|\mathcal{X}|$ through term $\mathbf{V}$. If $d-1 > \log(|\mathcal{X}|)/\log(1/\delta)$ and $\Gamma > 16\log(12/\delta)\sigma_{\max}^2 d^2(1+6\epsilon)$, then with probability $1 - \delta/2$,

$$
|\phi_x^\top(\widehat{\Psi}_\Gamma - \Psi^*)| \leq \sqrt{\frac{256\log(12/\delta)\sigma_{\max}^4 d^2(1+6\epsilon)}{\Gamma}}.
$$

Alternatively, if just $\Gamma > 16\log(12|\mathcal{X}|/\delta)\sigma_{\max}^2 d^2(1+6\epsilon)$, then

$$
|\phi_x^\top(\widehat{\Psi}_\Gamma - \Psi^*)| \leq \sqrt{\frac{256\log(12|\mathcal{X}|/\delta)\sigma_{\max}^4 d^2(1+6\epsilon)}{\Gamma}}. \tag{16}
$$

Since $\sigma_{\max}^2 \geq \sigma_x^2 \geq \sigma_{\min}^2$ for all $x \in \mathcal{X}$,

$$
|\phi_x^\top(\widehat{\Psi}_\Gamma - \Psi^*)| = |x^\top\widehat{\Sigma}_\Gamma x - x^\top\Sigma^* x| \geq |\min\left\{\max\left\{x^\top\widehat{\Sigma}_\Gamma x, \sigma_{\min}^2\right\}, \sigma_{\max}^2\right\} - \sigma_x^2|.
$$

In summary, if $\Gamma > 16\log(12|\mathcal{X}|/\delta)\sigma_{\max}^2 d^2(1+6\epsilon)$,

$$
\mathbb{P}\left(|\widehat{\sigma}_x^2 - \sigma_x^2| \leq \sqrt{\frac{C'\log(|\mathcal{X}|/\delta)\sigma_{\max}^4 d^2}{\Gamma}}\right) \geq 1 - \delta,
$$

where $C' = 256(1+6\epsilon)$. ∎

We now prove a simple lemma relating the additive bound on $|\widehat{\sigma}_x^2 - \sigma_x^2|$ to a multiplicative bound $1 - C_{\Gamma,\delta} < \sigma_x^2/\widehat{\sigma}_x^2 < 1 + C_{\Gamma,\delta}$.

**Lemma D.1.** Define $\widehat{\sigma}_x^2 = \min\left\{\max\left\{x^\top\widehat{\Sigma}_\Gamma x, \sigma_{\min}^2\right\}, \sigma_{\max}^2\right\}$. Letting $\sigma_{\min}^2 = \min_{x\in\mathcal{X}}\sigma_x^2$, $\kappa = \sigma_{\max}^2/\sigma_{\min}^2$, and

$$
\mathbb{P}\left(\forall x \in \mathcal{X}, 1 - C_{\Gamma,\delta} < \frac{\sigma_x^2}{\widehat{\sigma}_x^2} < 1 + C_{\Gamma,\delta}\right) \geq 1 - \delta/2, \text{ where } C_{\Gamma,\delta} = \sqrt{\frac{C'\log(|\mathcal{X}|/\delta)\kappa^2 d^2}{\Gamma}}.
$$

*Proof.* Using Lemma D.1, we know that with probability greater than $1 - \delta/2$ the following event holds.

$$|\sigma_x^2 - \widehat{\sigma}_x^2| \leq \sqrt{\frac{C'\log(|\mathcal{X}|/\delta)\sigma_{\max}^4 d^2}{\Gamma}} \Rightarrow$$

$$\widehat{\sigma}_x^2 - \sqrt{\frac{C'\log(|\mathcal{X}|/\delta)\sigma_{\max}^4 d^2}{\Gamma}} \leq \sigma_x^2 \leq \widehat{\sigma}_x^2 + \sqrt{\frac{C'\log(|\mathcal{X}|/\delta)\sigma_{\max}^4 d^2}{\Gamma}} \Rightarrow$$

$$\widehat{\sigma}_x^2 - \widehat{\sigma}_x^2\sqrt{\frac{C'\log(|\mathcal{X}|/\delta)\sigma_{\max}^4 d^2}{\widehat{\sigma}_x^4 \Gamma}} \leq \sigma_x^2 \leq \widehat{\sigma}_x^2 + \widehat{\sigma}_x^2\sqrt{\frac{C'\log(|\mathcal{X}|/\delta)\sigma_{\max}^4 d^2}{\widehat{\sigma}_x^4 \Gamma}} \Rightarrow$$

$$\widehat{\sigma}_x^2 - \widehat{\sigma}_x^2\sqrt{\frac{C'\log(|\mathcal{X}|/\delta)\sigma_{\max}^4 d^2}{\sigma_{\min}^4 \Gamma}} \leq \sigma_x^2 \leq \widehat{\sigma}_x^2 + \widehat{\sigma}_x^2\sqrt{\frac{C'\log(|\mathcal{X}|/\delta)\sigma_{\max}^4 d^2}{\sigma_{\min}^4 \Gamma}} \Rightarrow$$

$$1 - \sqrt{\frac{C'\log(|\mathcal{X}|/\delta)\kappa^2 d^2}{\Gamma}} \leq \frac{\sigma_x^2}{\widehat{\sigma}_x^2} \leq 1 + \sqrt{\frac{C'\log(|\mathcal{X}|/\delta)\kappa^2 d^2}{\Gamma}}.$$

∎

# E  Proofs of Best-Arm Identification

We divide the proof of Theorem 4.1 and 4.2 between Appendices E and F for ease of exposition. In Appendix E, we consider the transductive linear best-arm identification problem (BAI) in which we are interested in identifying $z^* = \max_{z \in \mathcal{Z}} z^\top \theta^*$ with probability $1 - \delta$ for $\delta \in (0, 1)$. In Appendix F, we consider the level set (LS) counterpart. Recall that for a set $\mathcal{V}$, let $P_\mathcal{V} := \{\lambda \in \mathbb{R}^{|\mathcal{V}|} : \sum_{v \in \mathcal{V}} \lambda_v = 1, \lambda_v \geq 0 \, \forall \, v\}$ denote distributions over $\mathcal{V}$ and $\mathcal{Y}(\mathcal{V}) := \{v' - v : v, v' \in \mathcal{V}, \, v \neq v'\}$ be an operator giving differences. We begin with the lower bound.

## E.1  Lower Bound

*Proof of Theorem 4.1 for transductive best-arm identification.* This proof is similar to the lower bound proofs of [35] and [12]. Let $\mathcal{C} := \{\widetilde{\theta} \in \mathbb{R}^d : \exists \, i \text{ s.t. } \widetilde{\theta}^\top(z^* - z_i) \leq 0\}$, i.e., $\widetilde{\theta} \in \mathcal{C}$ if and only if $z^*$ is not the best arm in the linear bandit instance $(\mathcal{X}, \mathcal{Z}, \widetilde{\theta})$. Additionally, label $K = |\mathcal{X}|$.

We first invoke Lemma 1 of [27]. Under a $\delta$-PAC strategy for finding the best arm for the bandit instance $(\mathcal{X}, \mathcal{Z}, \theta^*)$, let $T_i$ denote the random variable that is the number of times arm $i$ is pulled. Defining $\sigma_x^2 = x^\top \Sigma^* x$ for all $x \in \mathcal{X}$, we let $\nu_{\theta,i}$ denote the reward distribution of the $i$-th arm of $\mathcal{X}$, i.e., $\nu_{\theta,i} = \mathcal{N}(x_i^\top \theta, \sigma_{x_i}^2)$. For any $\widetilde{\theta} \in \mathcal{C}$ we know from [27] that

$$\sum_{i=1}^K \mathbb{E}(T_i) KL(\nu_{\theta^*,i} || \nu_{\widetilde{\theta},i}) \geq \log(1/2.4\delta).$$

Following the steps of [12], we find that

$$\sum_{i=1}^K \mathbb{E}(T_i) \geq \log(1/2.4\delta) \min_{\lambda \in P_\mathcal{X}} \max_{\widetilde{\theta} \in \mathcal{C}} \frac{1}{\sum_{i=1}^K \lambda_i KL(\nu_{\theta^*,i} || \nu_{\widetilde{\theta},i})}. \tag{17}$$

Label $\mathcal{Z} = \{z_1, z_2, \ldots, z_n\}$, where $z_1 = z^*$ without loss of generality. For $j \neq 1$, $\lambda \in P_\mathcal{X}$ and $\epsilon > 0$, define $A(\lambda) := \sum_{i=1}^K \lambda_i \frac{x_i x_i^\top}{\sigma_{x_i}^2}$, $w_j = z^* - z_j$ and

$$\theta_j(\epsilon, \lambda) = \theta^* - \frac{(w_j^\top \theta^* + \epsilon)A(\lambda)^{-1}w_j}{w_j^\top A(\lambda)^{-1}w_j}.$$

Note that $w_j^\top \theta_j(\epsilon, \lambda) = -\epsilon < 0$, implying that $\theta_j \in \mathcal{C}$. We find that the KL divergence between $\nu_{\theta,i}$ and $\nu_{\theta_j(\epsilon,\lambda),i}$ is given by:

$$KL(\nu_{\theta,i}||\nu_{\theta_j(\epsilon,\lambda),i}) = \frac{\left[x_i^\top \{\theta^* - \theta_j(\epsilon, \lambda)\}\right]^2}{2\sigma_{x_i}^2}$$

$$= w_j^\top A(\lambda)^{-1} \frac{(w_j^\top \theta^* + \epsilon)^2 \frac{x_i x_i^\top}{2\sigma_{x_i}^2}}{\left\{w_j^\top A(\lambda)^{-1} w_j\right\}^2} A(\lambda)^{-1} w_j.$$

Returning to Eq. 17,

$$\sum_{i=1}^K \mathbb{E}(T_i) \geq \log(1/2.4\delta) \min_{\lambda \in P_\mathcal{X}} \max_{\widetilde{\theta} \in \mathcal{C}} \frac{1}{\sum_{i=1}^K \lambda_i KL(\nu_{\theta^*,i}||\nu_{\widetilde{\theta},i})}$$

$$\geq \log(1/2.4\delta) \min_{\lambda \in P_\mathcal{X}} \max_{j=2,\cdots,m} \frac{1}{\sum_{i=1}^K \lambda_i KL(\nu_{\theta^*,i}||\nu_{\theta_j(\epsilon,\lambda),i})}$$

$$\geq \log(1/2.4\delta) \min_{\lambda \in P_\mathcal{X}} \max_{j=2,\cdots,m} \frac{\left\{w_j^\top A(\lambda)^{-1} w_j\right\}^2}{\sum_{i=1}^K (w_j^\top \theta^* + \epsilon)^2 w_j^\top A(\lambda)^{-1} \lambda_i \frac{x_i x_i^\top}{2\sigma_{x_i}^2} A(\lambda)^{-1} w_j}$$

$$\overset{(a)}{=} 2\log(1/2.4\delta) \min_{\lambda \in P_\mathcal{X}} \max_{j=2,\cdots,m} \frac{\left\{w_j^\top A(\lambda)^{-1} w_j\right\}^2}{(w_j^\top \theta^* + \epsilon)^2 w_j^\top A(\lambda)^{-1} \left(\sum_{i=1}^K \lambda_i \frac{x_i x_i^\top}{\sigma_{x_i}^2}\right) A(\lambda)^{-1} w_j}$$

$$= 2\log(1/2.4\delta) \min_{\lambda \in P_\mathcal{X}} \max_{j=2,\cdots,m} \frac{\left\{w_j^\top A(\lambda)^{-1} w_j\right\}^2}{(w_j^\top \theta^* + \epsilon)^2},$$

where $(a)$ uses the fact that $\sum_{i=1}^K \lambda_i \frac{x_i x_i^\top}{\sigma_{x_i}^2} = A(\lambda)$. Letting $\epsilon \to 0$ establishes the result. ∎

## E.2 Upper Bound

We now prove Theorem 4.2 by first establishing that Alg. 2 is $\delta$-PAC for the transductive linear bandits best-arm identification problem.

### E.2.1 Proof of Correctness

*Overview of Correctness Proof.* Lemma D.1 along with Lemmas E.1 through E.3 are combined to prove that Algorithm 2 is $\delta$-PAC for the best-arm identification problem. Lemma E.1 leverages Lemma D.1 and begins by establishing that in any round $\ell \in \mathbb{N}$, the estimation error for the difference in mean rewards between $z \in \mathcal{Z}$ and $z^* = \arg\max_{z \in \mathcal{Z}} z^\top \theta^*$ is bounded by $\epsilon_\ell = 2^{-\ell}$ with probability $1 - \delta$ for $\delta \in (0, 1)$. Lemma E.2 uses this result to prove that starting at round $\ell$, $z^* \in \mathcal{Z}_{\ell+1}$ with probability $1 - \delta$. This establishes that $z^*$ is not eliminated from the set of items we are considering at any step with high probability. Lastly, Lemma E.3 uses Lemma E.1 to show that Algorithm 2 will eventually identify $z^*$. This is done by proving that Alg. 2 eliminates items with mean rewards that are outside a shrinking radius (halves at every step of the algorithm) of the highest reward, $z^{*\top}\theta^*$. In other words, with probability $1 - \delta$ for $\delta \in (0, 1)$, we prove that items in $\mathcal{Z}_\ell$ with expected rewards that are $2\epsilon_\ell$ away from $z^{*\top}\theta^*$ will not be in $\mathcal{Z}_{\ell+1}$. Since $\epsilon_\ell \to 0$ as $\ell \to \infty$, we know that $\lim_{\ell \to \infty} \mathbb{P}(\mathcal{Z}_\ell = \{z^*\}) = 1 - \delta$.

**Lemma E.1.** With probability at least $1 - \delta$ for $\delta \in (0, 1)$,

$$|\langle z - z^*, \widehat{\theta}_\ell - \theta^* \rangle| \leq \epsilon_\ell$$

for all $\ell \in \mathbb{N}$ and $z \in \mathcal{Z}$.

*Proof.* Define $\sigma_x^2 = x^\top \Sigma^* x$, $\widehat{\theta}_\ell$ as the weighted least squares estimator constructed in Algorithm 2 at stage $\ell$, and events

$$\mathcal{E}_{z,\ell} = |\langle z - z^*, \widehat{\theta}_\ell - \theta^* \rangle| \leq \epsilon_\ell$$

and

$$\mathcal{J} = \left\{ \sigma_x^2 \le \widehat{\sigma}_x^2 \left(1 + C_{\Gamma,\delta}\right) = \frac{3}{2}\widehat{\sigma}_x^2, \ \forall x \in \mathcal{X} \right\}.$$

Label $n_\ell$ as the number of data points collected during stage $\ell$, $\Upsilon = \operatorname{diag}(\{\sigma_i^2\}_{i=1}^{n_\ell})$, $\widehat{\Upsilon} = \operatorname{diag}(\{\widehat{\sigma}_i^2\}_{i=1}^{n_\ell})$, $X_\ell$ as a matrix whose rows are composed of the arms $x_{i,\ell}$ drawn during round $\ell$, $Y_\ell$ as the vector of rewards drawn during round $\ell$, $A_\ell := \sum_{i=1}^{n_\ell} \widehat{\sigma}_i^{-2} x_{\ell,i} x_{\ell,i}^\top$, and $\widetilde{\Omega} = A_\ell^{-1} X_\ell^\top \widehat{\Upsilon}^{-1} \Upsilon \widehat{\Upsilon}^{-1} X_\ell A_\ell^{-1}$ as the weighted least squares estimator's covariance. We first establish that for $z \in \mathbb{R}^d$,

$$\mathbb{E}\left\{ z^\top (\widehat{\theta}_\ell - \theta^*) \right\} = z^\top \left\{ A_t^{-1} X_\ell \widehat{\Upsilon}^{-1} \mathbb{E}(Y_\ell) - \theta^* \right\} = z^\top \left( A_t^{-1} X_\ell \widehat{\Upsilon}^{-1} X_\ell \theta^* - \theta^* \right) = 0,$$

and

$$\mathbb{V}\left\{ z^\top (\widehat{\theta}_\ell - \theta^*) \right\} = z^\top \widetilde{\Omega} z.$$

Furthermore, defining $u^\top = z^\top A_\ell^{-1} X_\ell^\top \widehat{\Upsilon}^{-1/2} \in \mathbb{R}^{n_\ell}$ and conditioning on event $\mathcal{J}$,

$$
\begin{aligned}
z^\top \widetilde{\Omega} z &= z^\top A_\ell^{-1} X_\ell^\top \widehat{\Upsilon}^{-1} \Upsilon \widehat{\Upsilon}^{-1} X_\ell A_\ell^{-1} z \\
&= (z^\top A_\ell^{-1} X_\ell^\top \widehat{\Upsilon}^{-1/2}) \widehat{\Upsilon}^{-1/2} \Upsilon \widehat{\Upsilon}^{-1/2} (\widehat{\Upsilon}^{-1/2} X_\ell A_\ell^{-1} z) \\
&= u^\top \widehat{\Upsilon}^{-1/2} \Upsilon \widehat{\Upsilon}^{-1/2} u \\
&= \sum_{i=1}^{n_\ell} u_i^2 \frac{\sigma_{x_i}^2}{\widehat{\sigma}_i^2} \\
&\overset{(a)}{\le} \frac{3}{2} \sum_{i=1}^{n_\ell} u_i^2 \\
&= \frac{3}{2} z^\top A_\ell^{-1} X_\ell^\top \widehat{\Upsilon}^{-1} X_\ell A_\ell^{-1} z \\
&= \frac{3}{2} z^\top A_\ell^{-1} z,
\end{aligned}
$$

where $(a)$ is the result of event $\mathcal{J}$. Using this result and Proposition B.1, with probability at least $1 - \frac{\delta}{4\ell^2|\mathcal{Z}|}$,

$$
\begin{aligned}
\mathcal{E}_{z,\ell} = |\langle z - z^*, \widehat{\theta}_\ell - \theta \rangle| &\le ||z - z^*||_{\widetilde{\Omega}} \sqrt{2 \log(8\ell^2|\mathcal{Z}|/\delta)} \\
&\le \frac{3}{2} ||z - z^*||_{A_\ell^{-1}} \sqrt{2 \log(8\ell^2|\mathcal{Z}|/\delta)}.
\end{aligned}
$$

Conditioned on event $\mathcal{J}$, we have with probability $1 - \frac{\delta}{4\ell^2|\mathcal{Z}|}$,

$$
\begin{aligned}
|\langle z - z^*, \widehat{\theta}_\ell - \theta \rangle| &\le \frac{3}{2} ||z - z^*||_{\left(\sum_{x \in \mathcal{V}} \lceil \tau_\ell \lambda_{\ell,x} \rceil \frac{xx^\top}{\widehat{\sigma}_x^2}\right)^{-1}} \sqrt{2 \log(8\ell^2|\mathcal{Z}|/\delta)} \\
&\le \frac{3}{2} ||z - z^*||_{\left(\sum_{x \in \mathcal{X}} \lceil \tau_\ell \lambda_{\ell,x} \rceil \frac{xx^\top}{\widehat{\sigma}_x^2}\right)^{-1}} \sqrt{2 \log(8\ell^2|\mathcal{Z}|/\delta)} \\
&\le \frac{\frac{3}{2} ||z - z^*||_{\left(\sum_{x \in \mathcal{X}} \lambda_{\ell,x} \frac{xx^\top}{\widehat{\sigma}_x^2}\right)^{-1}}}{\sqrt{\tau_\ell}} \sqrt{2 \log(8\ell^2|\mathcal{Z}|/\delta)} \\
&= \sqrt{\frac{\frac{3}{2} ||z - z^*||^2_{\left(\sum_{x \in \mathcal{X}} \lambda_{\ell,x} \frac{xx^\top}{\widehat{\sigma}_x^2}\right)^{-1}}}{2 \left(\frac{3}{2}\right) \epsilon_\ell^{-2} q\left\{\mathcal{Y}(\mathcal{Z}_\ell)\right\} \log(8\ell^2|\mathcal{Z}|/\delta)}} \sqrt{2 \log(8\ell^2|\mathcal{Z}|/\delta)} \\
&\le \epsilon_\ell.
\end{aligned}
$$

Applying a union bound, we find that

$$\mathbb{P}\left\{\left(\cap_{\ell=1}^{\infty}\cap_{z\in\mathcal{Z}_{\ell}}\mathcal{E}_{z,\ell}|\mathcal{J}\right)\cap\mathcal{J}\right\} = 1 - \mathbb{P}\left\{\left(\cup_{\ell=1}^{\infty}\cup_{z\in\mathcal{Z}_{\ell}}\mathcal{E}_{z,\ell}^{c}|\mathcal{J}\right)\cup\mathcal{J}^{c}\right\}$$

$$\leq 1 - \left\{\sum_{\ell=1}^{\infty}\sum_{z\in\mathcal{Z}_{\ell}}\mathbb{P}(\mathcal{E}_{z,\ell}^{c}|\mathcal{J}) + \mathbb{P}(\mathcal{J}^{c})\right\}$$

$$= 1 - \left(\sum_{\ell=1}^{\infty}\sum_{z\in\mathcal{Z}_{\ell}}\frac{\delta}{4\ell^{2}|\mathcal{Z}|} + \frac{\delta}{2}\right)$$

$$\leq 1 - \delta,$$

which proves the result. ∎

Now we establish that, with probability $1 - \delta$ for $\delta \in (0,1)$, the best arm $z^*$ will not be eliminated from $\mathcal{Z}_{\ell}$ for any round $\ell \in \mathbb{N}$.

**Lemma E.2.** With probability at least $1 - \delta$ for $\delta \in (0,1)$, for any $\ell \in \mathbb{N}$ and $z' \in \mathcal{Z}_{\ell}$,

$$\langle z' - z^*, \widehat{\theta}_{\ell}\rangle \leq \epsilon_{\ell}.$$

*Proof.* We know that with probability $1 - \delta$ for all $\ell \in \mathbb{N}$ and $z' \in \mathcal{Z}_{\ell}$,

$$\langle z' - z^*, \widehat{\theta}_{\ell}\rangle = \langle z' - z^*, \widehat{\theta}_{\ell} - \theta^*\rangle + \langle z' - z^*, \theta^*\rangle$$

$$\leq \langle z' - z^*, \widehat{\theta}_{\ell} - \theta^*\rangle$$

$$\leq \epsilon_{\ell},$$

where the last inequality follows from Lemma E.1. ∎

Then, we establish that in round $\ell$, items in $\mathcal{Z}_{\ell}$ that have mean rewards farther than $2\epsilon_{\ell}$ from the highest reward will be eliminated.

**Lemma E.3.** With probability at least $1 - \delta$ for $\delta \in (0,1)$, for any $\ell \in \mathbb{N}$ and $z \in \mathcal{Z}_{\ell}$ such that $\langle z^* - z, \theta^*\rangle > 2\epsilon_{\ell}$,

$$\max_{z'\in\mathcal{Z}_{\ell}}\langle z' - z, \widehat{\theta}_{\ell}\rangle > \epsilon_{\ell}.$$

*Proof.* We know that with probability $1 - \delta$ for all $\ell \in \mathbb{N}$,

$$\max_{z'\in\mathcal{Z}_{\ell}}\langle z' - z, \widehat{\theta}_{\ell}\rangle \geq \langle z^* - z, \widehat{\theta}_{\ell}\rangle$$

$$= \langle z^* - z, \widehat{\theta}_{\ell} - \theta^*\rangle + \langle z^* - z, \theta^*\rangle$$

$$\overset{(a)}{>} -\epsilon_{\ell} + 2\epsilon_{\ell}$$

$$= \epsilon_{\ell},$$

where $(a)$ follows from Lemma E.1. ∎

#### E.2.2  Proof of Sample Complexity

*Overview of Sample Complexity Proof.* In this section, we prove an upper bound on the sample complexity of Algorithm 2 for transductive best-arm identification with linear bandits. We begin by establishing some basic facts about the partial ordering of Hermetian positive definite matrices in Lemmas E.4 and E.5. Let $A$ and $B$ be Hermitian positive definite matrices in $\mathbb{R}^{d\times d}$ and define $A \succeq B$ if $(A - B)$ is positive definite.

**Lemma E.4.** Let $A$ and $B$ be Hermetian Positive Definite matrices in $\mathbb{R}^{d\times d}$,

$$A \succeq B \iff I \succeq A^{-1/2}BA^{-1/2}.$$

*Proof.* For any $x \in \mathbb{R}^d$ we define $y = A^{1/2}x$ and

$$
\begin{aligned}
x^\top A x &= x^\top A^{1/2} A^{-1/2} A A^{-1/2} A^{1/2} x \\
&= y^\top I y \\
&= y^\top A^{-1/2} A A^{-1/2} y \\
&\geq x^\top B x \\
&= y^\top A^{-1/2} B A^{-1/2} y
\end{aligned}
$$

∎

**Lemma E.5.** Let $A$ and $B$ be Hermitian positive definite matrices in $\mathbb{R}^{d \times d}$,

$$
A \succeq B \iff B^{-1} \succeq A^{-1}.
$$

*Proof.* We use Lemma E.4 twice in the following proof.

$$
\begin{aligned}
A \succeq B &\iff I \succeq A^{-1/2} B A^{-1/2} \\
&\iff I \succeq B^{1/2} A^{-1/2} A^{-1/2} B^{1/2} \\
&\iff I \succeq B^{1/2} A^{-1} B^{1/2} \\
&\iff B^{-1} \succeq A^{-1}.
\end{aligned}
$$

where line 2 follows because $A^{-1/2} B A^{-1/2}$ is similar to $B^{1/2} A^{-1/2} A^{-1/2} B^{1/2}$ and both are Hermetian positive definite. ∎

We now use Lemmas E.4 and E.5 in our problem context. For $\mathcal{V} \subset \mathbb{R}^d$, define $g : (\mathcal{V}, P_{\mathcal{X}}) \to \mathbb{R}$ such that $g(v, \lambda) = \|v\|_{(\sum_x \lambda_x x x^\top / \widehat{\sigma}_x^2)^{-1}}$, $f : (\mathcal{V}, P_{\mathcal{X}}) \to \mathbb{R}$ such that $f(v, \lambda) = \|v\|_{(\sum_x \lambda_x x x^\top / \sigma_x^2)^{-1}}$ and

$$
q^*(\mathcal{V}) = \min_{\lambda \in P_{\mathcal{X}}} \max_{v \in \mathcal{V}} f(v, \lambda), \quad q(\mathcal{V}) = \min_{\lambda \in P_{\mathcal{X}}} \max_{v \in \mathcal{V}} g(v, \lambda).
$$

Note that the function, $q$, is also defined and leveraged in Algorithm 2. Lemma E.6 establishes a multiplicative bound of the form

$$
(1 - C_{\Gamma, \delta})\, g(v, \lambda) \leq f(v, \lambda) \leq (1 + C_{\Gamma, \delta})\, g(v, \lambda)
$$

using Lemmas D.1 and E.5.

**Lemma E.6.** Define $\widehat{\sigma}_x^2 = \min\left\{ \max\left\{ x^\top \widehat{\Sigma}_\Gamma x, \sigma_{\min}^2 \right\}, \sigma_{\max}^2 \right\}$. For a given $\lambda$, define

$$
\Upsilon = \mathrm{diag}(\{\sigma_x^2\}_{x \in \mathcal{X}}) \times \mathrm{diag}(\{1/\lambda_x\}_{x \in \mathcal{X}}), \quad \widehat{\Upsilon}^{-1} = \mathrm{diag}(\{\widehat{\sigma}_x^2\}_{x \in \mathcal{X}}) \times \mathrm{diag}(\{1/\lambda_x\}_{x \in \mathcal{X}}).
$$

Assume $C_{\Gamma, \delta} < 1$ and $v \in \mathbb{R}^d$, then with probability $1 - \delta/2$ for $\delta \in (0, 1)$ the following holds true:

$$
\begin{aligned}
(1 - C_{\Gamma, \delta})\, v^\top (X^\top \widehat{\Upsilon}^{-1} X)^{-1} v &\leq v^\top (X^\top \Upsilon^{-1} X)^{-1} v \\
&\leq (1 + C_{\Gamma, \delta})\, v^\top (X^\top \widehat{\Upsilon}^{-1} X)^{-1} v,
\end{aligned}
$$

or equivalently

$$
(1 - C_{\Gamma, \delta})\, g(v, \lambda) \leq f(v, \lambda) \leq (1 + C_{\Gamma, \delta})\, g(v, \lambda).
$$

*Proof.* Using Lemma D.1, we establish that

$$
\begin{aligned}
v^\top X^\top \widehat{\Upsilon}^{-1} X v &= v^\top X^\top \widehat{\Upsilon}^{-1} \Upsilon \Upsilon^{-1} X v \\
&= v^\top X^\top \Upsilon^{-1/2} \widehat{\Upsilon}^{-1} \Upsilon \Upsilon^{-1/2} X v \\
&\leq (1 + C_{\Gamma, \delta})\, v^\top X^\top \Upsilon^{-1} X v
\end{aligned}
$$

and

$$
\begin{aligned}
v^\top X^\top \widehat{\Upsilon}^{-1} X v &= v^\top X^\top \widehat{\Upsilon}^{-1} \Upsilon \Upsilon^{-1} X v \\
&= v^\top X^\top \Upsilon^{-1/2} \widehat{\Upsilon}^{-1} \Upsilon \Upsilon^{-1/2} X v \\
&\geq (1 - C_{\Gamma, \delta})\, v^\top X^\top \Upsilon^{-1} X v.
\end{aligned}
$$

Leveraging the partial ordering of Hermitian positive definite matrices and Lemma E.5, we can then represent these inequalities as

$$(1 - C_{\Gamma,\delta})\, X^\top \Upsilon^{-1} X \preceq X^\top \widehat{\Upsilon}^{-1} X \preceq (1 + C_{\Gamma,\delta})\, X^\top \Upsilon^{-1} X \iff$$

$$\frac{1}{1 - C_{\Gamma,\delta}} \left(X^\top \Upsilon^{-1} X\right)^{-1} \succeq \left(X^\top \widehat{\Upsilon}^{-1} X\right)^{-1} \succeq \frac{1}{1 + C_{\Gamma,\delta}} \left(X^\top \Upsilon^{-1} X\right)^{-1}.$$

This implies that

$$\left(X^\top \Upsilon^{-1} X\right)^{-1} \succeq (1 - C_{\Gamma,\delta}) \left(X^\top \widehat{\Upsilon}^{-1} X\right)^{-1}$$

and

$$\left(X^\top \Upsilon^{-1} X\right)^{-1} \preceq (1 + C_{\Gamma,\delta}) \left(X^\top \widehat{\Upsilon}^{-1} X\right)^{-1}.$$

By combining these inequalities we arrive at the result. ∎

We now want to use the multiplicative bound of Lemma E.6 to define the relationship between $q(\mathcal{V})$ and $q^*(\mathcal{V})$ and eventually relate the sample complexity upper bound of Alg. 2 to the lower bound established in Theorem 4.1. In order to accomplish this goal, we prove some general results concerning minimax problems in Lemmas E.7 and E.8.

**Lemma E.7.** For $c > 0$, $v \in \mathbb{R}^d$ and $\lambda \in P_\mathcal{X}$, if

$$(1 - c)g(v, \lambda) \le f(v, \lambda) \le (1 + c)g(v, \lambda),$$

then

$$(1 - c)\max_{v \in \mathcal{V}} g(v, \lambda) \le \max_{v \in \mathcal{V}} f(v, \lambda)$$
$$\le (1 + c)\max_{v \in \mathcal{V}} g(v, \lambda).$$

*Proof.* For a given $\lambda \in P_\mathcal{X}$, define $v^\dagger(\lambda) = \arg\max_{v \in \mathcal{V}} g(v, \lambda)$ and $v^*(\lambda) = \arg\max_{v \in \mathcal{V}} f(v, \lambda)$, then

$$(1 - c)g\left\{v^\dagger(\lambda), \lambda\right\} \le f\left\{v^\dagger(\lambda), \lambda\right\}$$
$$\le f\left\{v^*(\lambda), \lambda\right\}$$
$$\le (1 + c)g\left\{v^*(\lambda), \lambda\right\}$$
$$\le (1 + c)g\left\{v^\dagger(\lambda), \lambda\right\}.$$

∎

**Lemma E.8.** For $c > 0$, $v \in \mathbb{R}^d$ and $\lambda \in P_\mathcal{X}$, if

$$(1 - c)g(v, \lambda) \le f(v, \lambda) \le (1 + c)g(v, \lambda),$$

then

$$(1 - c)\min_{\lambda \in P_\mathcal{X}} \max_{v \in \mathcal{V}} g(v, \lambda) \le \min_{\lambda \in P_\mathcal{X}} \max_{v \in \mathcal{V}} f(v, \lambda)$$
$$\le (1 + c)\min_{\lambda \in P_\mathcal{X}} \max_{v \in \mathcal{V}} g(v, \lambda).$$

*Proof.* For a given $\lambda \in P_\mathcal{X}$, define $v^\dagger(\lambda) = \arg\max_{v \in \mathcal{V}} g(v, \lambda)$, $v^*(\lambda) = \arg\max_{v \in \mathcal{V}} f(v, \lambda)$, $\lambda^* = \arg\min_{\lambda \in P_\mathcal{X}} \max_{v \in \mathcal{V}} f(v, \lambda)$ and $\lambda^\dagger = \arg\min_{\lambda \in P_\mathcal{X}} \max_{v \in \mathcal{V}} g(v, \lambda)$, then

$$(1 - c)g\left\{v^\dagger(\lambda^\dagger), \lambda^\dagger\right\} \le (1 - c)g\left\{v^\dagger(\lambda^*), \lambda^*\right\}$$
$$\overset{(a)}{\le} f\left\{v^*(\lambda^*), \lambda^*\right\}$$
$$\le f\left\{v^*(\lambda^\dagger), \lambda^\dagger\right\}$$
$$\le (1 + c)g\left\{v^*(\lambda^\dagger), \lambda^\dagger\right\}$$
$$\overset{(b)}{\le} (1 + c)g\left\{v^\dagger(\lambda^\dagger), \lambda^\dagger\right\}$$

where lines $(a)$ and $(b)$ follows from Lemma E.7. ∎

Leveraging Lemma E.8, we bound $q^*(\mathcal{V})$ in terms of $q(\mathcal{V})$ in Lemma E.9.

**Lemma E.9.** With probability $1 - \delta/2$ for $\delta \in (0,1)$,
$$(1 - C_{\Gamma,\delta}) \, q(\mathcal{V}) \leq q^*(\mathcal{V}) \leq (1 + C_{\Gamma,\delta}) \, q(\mathcal{V}).$$

*Proof.* From Lemma E.6, we know that
$$(1 - C_{\Gamma,\delta}) \, g(v,\lambda) \leq f(v,\lambda) \leq (1 + C_{\Gamma,\delta}) \, g(v,\lambda).$$
According to Lemma E.8, this implies
$$(1 - C_{\Gamma,\delta}) \min_{\lambda \in P_\mathcal{X}} \max_{v \in \mathcal{V}} g(v,\lambda) \leq \min_{\lambda \in P_\mathcal{X}} \max_{v \in \mathcal{V}} f(v,\lambda) \leq (1 + C_{\Gamma,\delta}) \min_{\lambda \in P_\mathcal{X}} \max_{v \in \mathcal{V}} g(v,\lambda).$$
∎

In round $\ell$ of Algorithm 2, we sample arms in $\mathcal{X}$ according to a given design, $\widehat{\lambda}_\ell$. Although other rounding procedures are possible (see Proposition A.1), Theorem 4.2 assumes that we take $\lceil \tau_\ell \widehat{\lambda}_{\ell,x} \rceil$ samples of arm $x \in \mathcal{X}$ to accomplish this goal. This results in more than $\tau_\ell$ samples being taken in round $\ell$. We define the sparsity of any $\widehat{\lambda}_\ell$ design using Caratheodory's Theorem to motivate the number of extra samples needed by sampling in this manner.

**Lemma E.10.** [Appendix 8, [44]] In Algorithm 2, the support of $\widehat{\lambda}_\ell$ for all $\ell \in \mathbb{N}$ is $d(d+1)/2 + 1$.

*Proof.* Any design matrix in $\mathbb{R}^{d \times d}$ is symmetric and can consequently be expressed as a point in $\mathbb{R}^M$, $M = d(d+1)/2$. The design matrices leveraged in this paper belong to a convex hull of a subset of points since they are a convex combination of $xx^\top$ for $x \in \mathcal{X}$. According to Caratheodory's theorem, any point in the convex hull of any subset of points in $\mathbb{R}^D$ can be defined as a convex combination of $D + 1$ points. The optimal experimental designs in this paper can therefore be represented using only $M + 1$ points [44]. This sparsity holds regardless of the design's form and so applies to both the homoskedastic and heteroskedastic variance settings. ∎

Finally, we connect the sample complexity upper bound of Algorithm 2 to the lower bound for $\delta$-PAC algorithms established in Theorem 4.2 for transductive best-arm identification.

*Proof of Theorem 4.2 for transductive best-arm identification.* Assume that $\max_{z \in \mathcal{Z}_\ell} |\langle z - z^*, \theta^* \rangle| \leq 2$. Define $\mathcal{S}_\ell = \{ z \in \mathcal{Z}_\ell : \langle z^* - z, \theta^* \rangle \leq 4\epsilon_\ell \}$. Note that by assumption $\mathcal{Z} = \mathcal{Z}_1 = S_1$. Lemma E.3 implies that with probability at least $1 - \delta$ for $\delta \in (0,1)$, we know $\mathcal{Z}_\ell \subset \mathcal{S}_\ell$ for all $\ell \in \mathbb{N}$. This means that

$$q(\mathcal{Y}(\mathcal{Z}_\ell)) = \inf_{\lambda \in P_\mathcal{X}} \max_{z,z' \in \mathcal{Z}_\ell} ||z - z'||^2_{(\sum_{x \in \mathcal{X}} \lambda_x \frac{xx^\top}{\hat{\sigma}_x^2})^{-1}}$$

$$\leq \inf_{\lambda \in P_\mathcal{X}} \max_{z,z' \in \mathcal{S}_\ell} ||z - z'||^2_{(\sum_{x \in \mathcal{X}} \lambda_x \frac{xx^\top}{\hat{\sigma}_x^2})^{-1}}$$

$$= q(\mathcal{S}_\ell).$$

For $\ell \geq \lceil \log_2(4\Delta^{-1}) \rceil$, we know that $\mathcal{S}_\ell = \{z^*\}$. Thus, the sample size needed to identify $z^*$ is upper bounded as follows.

$$\sum_{\ell=1}^{\lceil \log_2(4\Delta^{-1}) \rceil} \sum_{x \in \mathcal{X}} \lceil \tau_\ell \widehat{\lambda}_{\ell,x} \rceil \overset{(a)}{=} \sum_{\ell=1}^{\lceil \log_2(4\Delta^{-1}) \rceil} \left\{ \frac{d(d+1)}{2} + \tau_\ell \right\}$$

$$= \sum_{\ell=1}^{\lceil \log_2(4\Delta^{-1}) \rceil} \left[ \frac{d(d+1)}{2} + 2 \left( \frac{3}{2} \right) \epsilon_\ell^{-2} q \{ \mathcal{Y}(\mathcal{Z}_\ell) \} \log(8\ell^2 |\mathcal{Z}|/\delta) \right]$$

$$\leq \frac{d(d+1)}{2} \lceil \log_2(4\Delta^{-1}) \rceil + \sum_{\ell=1}^{\lceil \log_2(4\Delta^{-1}) \rceil} 3 \epsilon_\ell^{-2} q(\mathcal{S}_\ell) \log(8\ell^2 |\mathcal{Z}|/\delta)$$

$$\leq \frac{d(d+1)}{2} \lceil \log_2(4\Delta^{-1}) \rceil$$

$$+ \quad 3 \log \left\{ \frac{8 \lceil \log_2(4\Delta^{-1}) \rceil^2 |\mathcal{Z}|}{\delta} \right\} \sum_{\ell=1}^{\lceil \log_2(4\Delta^{-1}) \rceil} 2^{2\ell} q(\mathcal{S}_\ell),$$

where $(a)$ follows from Lemma E.10.

We relate this quantity to the lower bound in Theorem 4.1 below. With probability $1 - \delta$,

$$\psi^*_{\text{BAI}} = \inf_{\lambda \in P_{\mathcal{X}}} \max_{z \in \mathcal{Z} \setminus \{z^*\}} \frac{||z^* - z||^2_{\left(\sum_{x \in \mathcal{X}} \lambda_x \frac{xx^\top}{\sigma_x^2}\right)^{-1}}}{\{(z^* - z)^\top \theta^*\}^2}$$

$$= \inf_{\lambda \in P_{\mathcal{X}}} \max_{\ell \leq \lceil \log_2(4\Delta^{-1}) \rceil} \max_{z \in \mathcal{S}_\ell} \frac{||z^* - z||^2_{\left(\sum_{x \in \mathcal{X}} \lambda_x \frac{xx^\top}{\sigma_x^2}\right)^{-1}}}{\{(z^* - z)^\top \theta^*\}^2}$$

$$\geq \inf_{\lambda \in P_{\mathcal{X}}} \max_{\ell \leq \lceil \log_2(4\Delta^{-1}) \rceil} \max_{z \in \mathcal{S}_\ell} \frac{||z^* - z||^2_{\left(\sum_{x \in \mathcal{X}} \lambda_x \frac{xx^\top}{\sigma_x^2}\right)^{-1}}}{(4 \times 2^{-\ell})^2}$$

$$\overset{(a)}{\geq} \frac{1}{\lceil \log_2(4\Delta^{-1}) \rceil} \inf_{\lambda \in P_{\mathcal{X}}} \sum_{\ell=1}^{\lceil \log_2(4\Delta^{-1}) \rceil} \max_{z \in \mathcal{S}_\ell} \frac{||z^* - z||^2_{\left(\sum_{x \in \mathcal{X}} \lambda_x \frac{xx^\top}{\sigma_x^2}\right)^{-1}}}{(4 \times 2^{-\ell})^2}$$

$$\overset{(b)}{\geq} \frac{1}{16 \lceil \log_2(4\Delta^{-1}) \rceil} \sum_{\ell=1}^{\lceil \log_2(4\Delta^{-1}) \rceil} 2^{2\ell} \inf_{\lambda \in P_{\mathcal{X}}} \max_{z \in \mathcal{S}_\ell} ||z^* - z||^2_{\left(\sum_{x \in \mathcal{X}} \lambda_x \frac{xx^\top}{\sigma_x^2}\right)^{-1}}$$

$$\overset{(c)}{\geq} \frac{1}{64 \lceil \log_2(4\Delta^{-1}) \rceil} \sum_{\ell=1}^{\lceil \log_2(4\Delta^{-1}) \rceil} 2^{2\ell} \inf_{\lambda \in P_{\mathcal{X}}} \max_{z, z' \in \mathcal{S}_\ell} ||z' - z||^2_{\left(\sum_{x \in \mathcal{X}} \lambda_x \frac{xx^\top}{\sigma_x^2}\right)^{-1}}$$

$$\overset{(d)}{\geq} \frac{1}{64 \lceil \log_2(4\Delta^{-1}) \rceil} \sum_{\ell=1}^{\lceil \log_2(4\Delta^{-1}) \rceil} 2^{2\ell} \inf_{\lambda \in P_{\mathcal{X}}} \max_{z, z' \in \mathcal{S}_\ell} \frac{1}{2} ||z' - z||^2_{\left(\sum_{x \in \mathcal{X}} \lambda_x \frac{xx^\top}{\hat{\sigma}_x^2}\right)^{-1}}$$

$$\geq \frac{1}{128 \lceil \log_2(4\Delta^{-1}) \rceil} \sum_{\ell=1}^{\lceil \log_2(4\Delta^{-1}) \rceil} 2^{2\ell} \inf_{\lambda \in P_{\mathcal{X}}} \max_{z, z' \in \mathcal{S}_\ell} ||z' - z||^2_{\left(\sum_{x \in \mathcal{X}} \lambda_x \frac{xx^\top}{\hat{\sigma}_x^2}\right)^{-1}}$$

$$= \frac{1}{128 \lceil \log_2(4\Delta^{-1}) \rceil} \sum_{\ell=1}^{\lceil \log_2(4\Delta^{-1}) \rceil} 2^{2\ell} q(\mathcal{S}_\ell)$$

here $(a)$ follows from the fact that the maximum of a set of numbers is always greater than the average and $(b)$ by the fact that the minimum of a sum is greater than the sum of the minimums. We used the fact that

$$\max_{z, z' \in \mathcal{S}_\ell} ||z' - z||^2_{\left(\sum_{x \in \mathcal{X}} \lambda_x \frac{xx^\top}{\sigma_x^2}\right)^{-1}} \leq 4 \max_{z \in \mathcal{S}_\ell} ||z^* - z||^2_{\left(\sum_{x \in \mathcal{X}} \lambda_x \frac{xx^\top}{\sigma_x^2}\right)^{-1}}$$

by the triangle inequality in $(c)$. Finally, $(d)$ follows from Lemma E.9.

Consequently, putting the previous pieces together we find that the sample complexity can be upper bounded as

$$\sum_{\ell=1}^{\lceil \log_2(4\Delta^{-1}) \rceil} \sum_{x \in \mathcal{X}} \lceil \tau_\ell \hat{\lambda}_{\ell, x} \rceil \leq \frac{d(d+1)}{2} \lceil \log_2(4\Delta^{-1}) \rceil$$

$$+ \quad 3 \log \left\{ \frac{8 \lceil \log_2(4\Delta^{-1}) \rceil^2 |\mathcal{Z}|}{\delta} \right\} \sum_{\ell=1}^{\lceil \log_2(4\Delta^{-1}) \rceil} 2^{2\ell} q(\mathcal{S}_\ell)$$

$$\leq \frac{d(d+1)}{2} \lceil \log_2(4\Delta^{-1}) \rceil$$

$$+ \quad 384 \lceil \log_2(4\Delta^{-1}) \rceil \log \left\{ \frac{8 \lceil \log_2(4\Delta^{-1}) \rceil^2 |\mathcal{Z}|}{\delta} \right\} \psi^*_{\text{BAI}},$$

where the dependency on $d(d+1)/2$ can reduced to $d$ by Proposition A.1. This proves the result. $\blacksquare$

# F   Proofs of Level Set Identification

This appendix contains the proofs for the results on transductive linear bandit level set identification. In this problem, we are interested in identifying the set $G_\alpha(\theta^*) = \{z \in \mathcal{Z} : z^\top \theta^* > \alpha\}$, i.e., the items with mean rewards that exceed threshold $\alpha$. We prove the lower bound on the sample complexity from Theorem 4.1 in App. F.1. The proof of correctness and the sample complexity upper bound for Alg. 2 are presented in App. F.2. These proofs resemble the transductive linear bandit identification proofs with slight modifications accounting for the the problem structure. Recalling $B_\alpha(\theta^*) = \{z \in \mathcal{Z} : z^\top \theta^* < \alpha\}$, we assume that $\mathcal{Z} = G_\alpha(\theta^*) \cup B_\alpha(\theta^*)$, which is equivalent to the condition $\{z \in \mathcal{Z} : z^\top \theta^* = \alpha\} = \emptyset$. Let $\Delta = \min_{z \in \mathcal{Z}} |z^\top \theta^* - \alpha|$ denote the minimum absolute gap for the threshold $\alpha$.

## F.1   Lower Bound

*Proof of Theorem 4.1 for transductive level set identification.* Label $\mathcal{X} \subset \mathbb{R}^d$ as the measurement vectors such that $K = |\mathcal{X}|$ and $\mathcal{Z}$ as the decision item set. This proof is similar to the lower bound proofs of [35] and [12]. Let $\mathcal{C}$ be the set of alternatives such that for $\widetilde{\theta} \in \mathcal{C}$, $G_\alpha(\theta^*) \neq G_\alpha(\widetilde{\theta})$. This set of alternatives is decomposed by [35] as,

$$\mathcal{C} = \left( \cup_{z \in G_\alpha(\theta^*)} \{\widetilde{\theta} : z^\top \widetilde{\theta} < \alpha\} \right) \cup \left( \cup_{z \notin G_\alpha(\theta^*)} \{\widetilde{\theta} : z^\top \widetilde{\theta} > \alpha\} \right).$$

We now recall Lemma 1 of [27]. Under a $\delta$-PAC strategy for bandit instance $(\mathcal{X}, \mathcal{Z}, \theta^*)$, let $T_i$ denote the random variable that is the number of times arm $i$ is pulled. Defining $\sigma_x^2 = x^\top \Sigma^* x$ for all $x \in \mathcal{X}$, we denote $\nu_{\theta,i}$ as the reward distribution of the $i$-th arm of $\mathcal{X}$, i.e., $\nu_{\theta,i} = \mathcal{N}\left(x_i^\top \theta, \sigma_{x_i}^2\right)$. For any $\widetilde{\theta} \in \mathcal{C}$, we know from [27] that

$$\sum_{i=1}^{K} \mathbb{E}(T_i) KL(\nu_{\theta^*,i} || \nu_{\widetilde{\theta},i}) \geq \log(1/2.4\delta).$$

Following the steps of [12], we find that

$$\sum_{i=1}^{K} \mathbb{E}(T_i) \geq \log(1/2.4\delta) \min_{\lambda \in P_\mathcal{X}} \max_{\widetilde{\theta} \in \mathcal{C}} \frac{1}{\sum_{i=1}^{K} \lambda_i KL(\nu_{\theta^*,i} || \nu_{\widetilde{\theta},i})}. \tag{18}$$

For $\lambda \in P_\mathcal{X}$, define $A(\lambda) := \sum_{i=1}^{K} \lambda_i \frac{x_i x_i^\top}{\sigma_{x_i}^2}$. For $\epsilon > 0$ and $z \in G_\alpha(\theta^*)$, define

$$\theta_z(\epsilon, \lambda) := \theta^* - \frac{(\theta^{*\top} z - \alpha + \epsilon) A(\lambda)^{-1} z}{\|z\|_{A(\lambda)^{-1}}^2},$$

where $z^\top \theta_z(\epsilon, \lambda) = \alpha - \epsilon$, implying that $\theta_z(\epsilon, \lambda) \in \mathcal{C}$. For $z \in G_\alpha^c(\theta^*)$, define

$$\theta_z(\epsilon, \lambda) := \theta^* - \frac{(\theta^{*\top} z - \alpha - \epsilon) A(\lambda)^{-1} z}{\|z\|_{A(\lambda)^{-1}}^2},$$

where $z^\top \theta_z(\epsilon, \lambda) = \alpha + \epsilon$, implying that $\theta_z(\epsilon, \lambda) \in \mathcal{C}$.

Defining $\epsilon_z = \epsilon$ if $z \in G_\alpha(\theta^*)$ and $\epsilon_z = -\epsilon$ if $z \in G_\alpha^c(\theta^*)$, we find that the KL divergence between $\nu_{\theta,i}$ and $\nu_{\theta_z(\epsilon,\lambda),i}$ is given by:

$$
\begin{aligned}
KL(\nu_{\theta,i} || \nu_{\theta_z(\epsilon,\lambda),i}) &= \frac{\left[x_i^\top \{\theta^* - \theta_z(\epsilon, \lambda)\}\right]^2}{2\sigma_{x_i}^2} \\
&= z^\top A(\lambda)^{-1} \frac{(z^\top \theta^* - \alpha + \epsilon_z)^2 \frac{x_i x_i^\top}{2\sigma_{x_i}^2}}{\{z^\top A(\lambda)^{-1} z\}^2} A(\lambda)^{-1} z.
\end{aligned}
$$

Returning to Eq. 18,

$$
\sum_{i=1}^{K} \mathbb{E}(T_i) \geq \log(1/2.4\delta) \min_{\lambda \in P_{\mathcal{X}}} \max_{\widetilde{\theta} \in \mathcal{C}} \frac{1}{\sum_{i=1}^{K} \lambda_i KL(\nu_{\theta^*,i} || \nu_{\widetilde{\theta},i})}
$$

$$
\geq \log(1/2.4\delta) \min_{\lambda \in P_{\mathcal{X}}} \max_{z \in \mathcal{Z}} \frac{1}{\sum_{i=1}^{K} \lambda_i KL(\nu_{\theta^*,i} || \nu_{\theta_z(\epsilon_z,\lambda),i})}
$$

$$
\geq \log(1/2.4\delta) \min_{\lambda \in P_{\mathcal{X}}} \max_{z \in \mathcal{Z}} \frac{\left\{ z^\top A(\lambda)^{-1} z \right\}^2}{\sum_{i=1}^{K} (z^\top \theta^* - \alpha + \epsilon_z)^2 z^\top A(\lambda)^{-1} \lambda_i \frac{x_i x_i^\top}{2\sigma_{x_i}^2} A(\lambda)^{-1} z}
$$

$$
\overset{(a)}{=} 2 \log(1/2.4\delta) \min_{\lambda \in P_{\mathcal{X}}} \max_{z \in \mathcal{Z}} \frac{\left\{ z^\top A(\lambda)^{-1} z \right\}^2}{(z^\top \theta^* - \alpha + \epsilon_z)^2 z^\top A(\lambda)^{-1} \left( \sum_{i=1}^{K} \lambda_i \frac{x_i x_i^\top}{\sigma_{x_i}^2} \right) A(\lambda)^{-1} z}
$$

$$
= 2 \log(1/2.4\delta) \min_{\lambda \in P_{\mathcal{X}}} \max_{z \in \mathcal{Z}} \frac{z^\top A(\lambda)^{-1} z}{(z^\top \theta^* - \alpha + \epsilon_z)^2},
$$

where $(a)$ uses the fact that $\sum_{i=1}^{K} \lambda_i \frac{x_i x_i^\top}{\sigma_{x_i}^2} = A(\lambda)$. Letting $\epsilon_z \to 0$ establishes the result. ∎

## F.2   Upper Bound

We now prove the upper bound on the sample complexity of Alg. 2 for level set estimation. This proof follows similarly to the upper bound on the sample complexity of Alg. 2 for best-arm identification shown in App. E.2. Note that in each round $\ell \in \mathbb{N}$, Alg. 2 maintains a set $\mathcal{Z}_\ell$ of undecided items, a set $\mathcal{G}_\ell$ of items estimated to have value exceeding the threshold $\alpha$, and a set $\mathcal{B}_\ell$ of items estimated to have value falling below the threshold $\alpha$.

### F.2.1   Proof of Correctness

To begin, we show that that for all rounds $\ell \in \mathbb{N}$ of Alg. 2, the error in the estimates of $z^\top \theta^*$ for all $z \in \mathcal{Z}$ are bounded by $\epsilon_\ell$ with probability at least $1 - \delta$ for $\delta \in (0,1)$.

**Lemma F.1.** For all $\ell \in \mathbb{N}$ and $z \in \mathcal{Z}$, with probability at least $1 - \delta$ for $\delta \in (0,1)$,

$$
|\langle z, \widehat{\theta}_\ell - \theta \rangle| \leq \epsilon_\ell.
$$

*Proof.* This proof follows identically to Lemma E.1 by replacing $z^*$ with the zero vector. ∎

We now apply the preceding result to show that with probability $1 - \delta$ for $\delta \in (0,1)$, Alg. 2 does not place items in the wrong category.

**Lemma F.2.** With probability at least $1 - \delta$ for $\delta \in (0,1)$, for any $z \in \mathcal{Z}_\ell$, $\ell \in \mathbb{N}$, such that $z^\top \theta^* > \alpha$,

$$
\langle z, \widehat{\theta}_\ell \rangle + \epsilon_\ell > \alpha,
$$

and for any $z \in \mathcal{Z}_\ell$ such that $z^\top \theta^* < \alpha$,

$$
\langle z, \widehat{\theta}_\ell \rangle - \epsilon_\ell < \alpha.
$$

*Proof.* Consider $z \in G_\alpha$ so that $z^\top \theta^* > \alpha$. Using Lemma F.1, with probability at least $1 - \delta$ for $\delta \in (0,1)$ we have

$$
\langle z, \widehat{\theta}_\ell \rangle + \epsilon_\ell = \langle z, \widehat{\theta}_\ell - \theta^* \rangle + \epsilon_\ell + \langle z, \theta^* \rangle
$$
$$
> \langle z, \widehat{\theta}_\ell - \theta^* \rangle + \epsilon_\ell + \alpha
$$
$$
\geq \alpha.
$$

By the procedure of Alg. 2, this guarantees $\mathcal{G}_{\ell+1} \subseteq G_\alpha$.

Now, consider $z \in B_\alpha$ so that $z^\top \theta^* < \alpha$. Using Lemma F.1, with probability at least $1 - \delta$ for $\delta \in (0, 1)$ we have

$$
\begin{aligned}
\langle z, \widehat{\theta}_\ell \rangle - \epsilon_\ell = \langle z, \widehat{\theta}_\ell - \theta^* \rangle - \epsilon_\ell + \langle z, \theta^* \rangle \\
< \langle z, \widehat{\theta}_\ell - \theta^* \rangle - \epsilon_\ell + \alpha \\
\leq \alpha.
\end{aligned}
$$

By the procedure of Alg. 2, this guarantees $\mathcal{B}_{\ell+1} \subseteq B_\alpha$. ∎

We now show that any $z \in \mathcal{Z}$ such that $|z^\top \theta^* - \alpha| > 2\epsilon_\ell$ is removed from the active set, $\mathcal{Z}_\ell$, in round $\ell \in \mathbb{N}$ of Alg. 2 with probability at least $1 - \delta$ for $\delta \in (0, 1)$. Since $\epsilon_\ell \to 0$, this proves that we will eventually identify $G_\alpha$ correctly.

**Lemma F.3.** With probability at least $1 - \delta$ for $\delta \in (0, 1)$, for any $\ell \in \mathbb{N}$ and $z \in \mathcal{Z}_\ell$ such that $|\langle z, \theta^* \rangle - \alpha| > 2\epsilon_\ell$,

$$
|\langle z, \widehat{\theta}_\ell \rangle - \alpha| > \epsilon_\ell.
$$

*Proof.* Set $\delta \in (0, 1)$. Let us begin by considering any $z \in \mathcal{Z}_\ell$ such that $\langle z, \theta^* \rangle - \alpha > 2\epsilon_\ell$. Using Lemma F.1, with probability at least $1 - \delta$,

$$
\begin{aligned}
\langle z, \widehat{\theta}_\ell \rangle - \epsilon_\ell = \langle z, \widehat{\theta}_\ell - \theta^* \rangle + \langle z, \theta^* \rangle - \epsilon_\ell \\
\geq -\epsilon_\ell + \langle z, \theta^* \rangle - \epsilon_\ell \\
> \alpha.
\end{aligned}
$$

Now, consider any $z \in \mathcal{Z}_\ell$ such that $\langle z, \theta^* \rangle - \alpha < -2\epsilon_\ell$. Using Lemma F.1, with probability at least $1 - \delta$,

$$
\begin{aligned}
\langle z, \widehat{\theta}_\ell \rangle + \epsilon_\ell = \langle z, \widehat{\theta}_\ell - \theta^* \rangle + \langle z, \theta^* \rangle + \epsilon_\ell \\
\leq \epsilon_\ell + \langle z, \theta^* \rangle + \epsilon_\ell \\
< \alpha.
\end{aligned}
$$

Hence, for $z \in \mathcal{Z}_\ell$ such that $|\langle z, \theta^* \rangle - \alpha| > 2\epsilon_\ell$, we have that $|\langle z, \widehat{\theta}_\ell \rangle - \epsilon_\ell| > \alpha$ with probability at least $1 - \delta$. ∎

By the elimination conditions of Alg. 2, this guarantees that the $z \in \mathcal{Z}_\ell$ such that $|\langle z, \theta^* \rangle - \alpha| > 2\epsilon_\ell$ are not included in $\mathcal{Z}_{\ell+1}$.

### F.2.2 Proof of Upper Bound on Sample Complexity

Now we prove the upper bound on the sample complexity of Alg. 2 for the transductive level set identification problem.

*Proof of Theorem 4.2 for transductive level set identification.* Set $\delta \in (0, 1)$. Assume that $\max_{z \in \mathcal{Z}_\ell} |\langle z, \theta^* \rangle - \alpha| \leq 2$. Define $\mathcal{S}_\ell = \{z \in \mathcal{Z}_\ell : |\langle z, \theta^* \rangle - \alpha| \leq 4\epsilon_\ell\}$ for $\ell \in \mathbb{N}$. Note that by assumption $\mathcal{Z} = \mathcal{S}_1 = \mathcal{Z}_1$. Lemma F.3 implies that with probability at least $1 - \delta$, we know $\mathcal{Z}_\ell \subset \mathcal{S}_\ell$ for all $\ell \in \mathbb{N}$. This means

$$
\begin{aligned}
q(\mathcal{Z}_\ell) = \inf_{\lambda \in P_\mathcal{X}} \max_{z \in \mathcal{Z}_\ell} ||z||^2_{\left(\sum_{x \in \mathcal{X}} \lambda_x \frac{xx^\top}{\widehat{\sigma}_x^2}\right)^{-1}} \\
\leq \inf_{\lambda \in P_\mathcal{X}} \max_{z \in \mathcal{S}_\ell} ||z||^2_{\left(\sum_{x \in \mathcal{X}} \lambda_x \frac{xx^\top}{\widehat{\sigma}_x^2}\right)^{-1}} \\
= q(\mathcal{S}_\ell).
\end{aligned}
$$

For $\ell \geq \lceil \log_2(4\Delta^{-1}) \rceil$, we know that $\mathcal{S}_\ell = \emptyset$, thus the sample complexity to identify the set $G_\alpha$ is upper bounded as follows.

$$\sum_{\ell=1}^{\lceil \log_2(4\Delta^{-1}) \rceil} \sum_{x \in \mathcal{X}} \lceil \tau_\ell \widehat{\lambda}_{\ell,x} \rceil \overset{(a)}{=} \sum_{\ell=1}^{\lceil \log_2(4\Delta^{-1}) \rceil} \left\{ \frac{d(d+1)}{2} + \tau_\ell \right\}$$

$$= \sum_{\ell=1}^{\lceil \log_2(4\Delta^{-1}) \rceil} \left\{ \frac{d(d+1)}{2} + 2\left(\frac{3}{2}\right) \epsilon_\ell^{-2} q(\mathcal{Z}_\ell) \log(8\ell^2 |\mathcal{Z}|/\delta) \right\}$$

$$\leq \frac{d(d+1)}{2} \lceil \log_2(4\Delta^{-1}) \rceil + \sum_{\ell=1}^{\lceil \log_2(4\Delta^{-1}) \rceil} 3\epsilon_\ell^{-2} q(\mathcal{S}_\ell) \log(8\ell^2 |\mathcal{Z}|/\delta)$$

$$\leq \frac{d(d+1)}{2} \lceil \log_2(4\Delta^{-1}) \rceil +$$

$$3\log\left\{ \frac{8\lceil \log_2(4\Delta^{-1}) \rceil^2 |\mathcal{Z}|}{\delta} \right\} \sum_{\ell=1}^{\lceil \log_2(4\Delta^{-1}) \rceil} 2^{2\ell} q(\mathcal{S}_\ell),$$

where $(a)$ follows from Lemma E.10. We now relate this quantity to the lower bound and explain below. With probability $1 - \delta$,

$$\psi_{\text{LS}}^* = \inf_{\lambda \in P_\mathcal{X}} \max_{z \in \mathcal{Z}} \frac{||z||^2_{\left(\sum_{x \in \mathcal{X}} \lambda_x \frac{xx^\top}{\sigma_x^2}\right)^{-1}}}{(z^\top \theta^* - \alpha)^2}$$

$$= \inf_{\lambda \in P_\mathcal{X}} \max_{\ell \leq \lceil \log_2(4\Delta^{-1}) \rceil} \max_{z \in \mathcal{S}_\ell} \frac{||z||^2_{\left(\sum_{x \in \mathcal{X}} \lambda_x \frac{xx^\top}{\sigma_x^2}\right)^{-1}}}{(z^\top \theta^* - \alpha)^2}$$

$$\geq \inf_{\lambda \in P_\mathcal{X}} \max_{\ell \leq \lceil \log_2(4\Delta^{-1}) \rceil} \max_{z \in \mathcal{S}_\ell} \frac{||z||^2_{\left(\sum_{x \in \mathcal{X}} \lambda_x \frac{xx^\top}{\sigma_x^2}\right)^{-1}}}{(4 \times 2^{-\ell})^2}$$

$$\overset{(a)}{\geq} \frac{1}{16\lceil \log_2(4\Delta^{-1}) \rceil} \inf_{\lambda \in P_\mathcal{X}} \sum_{\ell=1}^{\lceil \log_2(4\Delta^{-1}) \rceil} 2^{2\ell} \max_{z \in \mathcal{S}_\ell} ||z||^2_{\left(\sum_{x \in \mathcal{X}} \lambda_x \frac{xx^\top}{\sigma_x^2}\right)^{-1}}$$

$$\overset{(b)}{\geq} \frac{1}{16\lceil \log_2(4\Delta^{-1}) \rceil} \sum_{\ell=1}^{\lceil \log_2(4\Delta^{-1}) \rceil} 2^{2\ell} \inf_{\lambda \in P_\mathcal{X}} \max_{z \in \mathcal{S}_\ell} ||z||^2_{\left(\sum_{x \in \mathcal{X}} \lambda_x \frac{xx^\top}{\sigma_x^2}\right)^{-1}}$$

$$\overset{(c)}{\geq} \frac{1}{32\lceil \log_2(4\Delta^{-1}) \rceil} \sum_{\ell=1}^{\lceil \log_2(4\Delta^{-1}) \rceil} 2^{2\ell} \inf_{\lambda \in P_\mathcal{X}} \max_{z \in \mathcal{S}_\ell} ||z||^2_{\left(\sum_{x \in \mathcal{X}} \lambda_x \frac{xx^\top}{\widehat{\sigma}_x^2}\right)^{-1}}$$

$$= \frac{1}{32\lceil \log_2(4\Delta^{-1}) \rceil} \sum_{\ell=1}^{\lceil \log_2(4\Delta^{-1}) \rceil} 2^{2\ell} q(\mathcal{S}_\ell).$$

$\blacksquare$

where $(a)$ follows from the fact that the maximum of a set of numbers is always greater than the average, $(b)$ by the fact that the minimum of a sum is greater than the sum of the minimums, and $(c)$ from Lemma E.9. Consequently, putting the previous pieces together we find that the sample

complexity can be upper bounded as

$$
\sum_{\ell=1}^{\lceil \log_2(4\Delta^{-1}) \rceil} \sum_{x \in \mathcal{X}} \lceil \tau_\ell \widehat{\lambda}_{\ell,x} \rceil \leq \frac{d(d+1)}{2} \lceil \log_2(4\Delta^{-1}) \rceil
$$

$$
+ \quad 3 \log \left\{ \frac{8 \lceil \log_2(4\Delta^{-1}) \rceil^2 |\mathcal{Z}|}{\delta} \right\} \sum_{\ell=1}^{\lceil \log_2(4\Delta^{-1}) \rceil} 2^{2\ell} q(\mathcal{S}_\ell)
$$

$$
\leq \frac{d(d+1)}{2} \lceil \log_2(4\Delta^{-1}) \rceil
$$

$$
+ \quad 96 \lceil \log_2(4\Delta^{-1}) \rceil \log \left\{ \frac{8 \lceil \log_2(4\Delta^{-1}) \rceil^2 |\mathcal{Z}|}{\delta} \right\} \psi_{\mathrm{LS}}^*,
$$

where the dependency on $d(d+1)/2$ can reduced to $d$ by Proposition A.1. This proves the result.

## G  Comparing Identification Lower Bounds

In this section, we use the general notation for the level set and best-arm identification problems developed in Section 4. Define

$$
\rho_{\mathrm{OBJ}}^* := \sigma_{\max}^2 \min_{\lambda \in P_{\mathcal{X}}} \max_{h \in \mathcal{H}_{\mathrm{OBJ}}, q \in \mathcal{Q}_{\mathrm{OBJ}}} \frac{\|h-q\|_{(\sum_{x \in \mathcal{X}} \lambda_x x x^\top)^{-1}}^2}{\{(h-q)^\top \theta^* - b_{\mathrm{OBJ}}\}^2},
$$

which is the lower bound for both BAI and LS by [12] and [35] in the homoskedastic case when upper bounding the variance.

**Lemma G.1.** In the same setting as Theorem 4.1,

$$
\sigma_{\min}^2 \cdot \rho_{\mathrm{OBJ}}^* \leq \psi_{\mathrm{OBJ}}^* \leq \sigma_{\max}^2 \cdot \rho_{\mathrm{OBJ}}^*,
$$

and both the upper and lower bounds are tight.

*Proof.* Consider semidefinite matrices $A$, $B$, and $C$ in $\mathbb{R}^{d \times d}$ such that $A \preceq B \preceq C$. Then $C^{-1} \preceq B^{-1} \preceq A^{-1}$. Hence, for $x \in \mathbb{R}^d$, $\|x\|_{C^{-1}} \leq \|x\|_{B^{-1}} \leq \|x\|_{A^{-1}}$. Defining

$$
A(\lambda) := \sum_{i=1}^K \lambda_i \frac{x_i x_i^\top}{\sigma_{x_i}^2},
$$

we note that

$$
\frac{1}{\sigma_{\max}^2} \sum_{i=1}^K \lambda_i x_i x_i^\top \preceq A(\lambda) \preceq \frac{1}{\sigma_{\min}^2} \sum_{i=1}^K \lambda_i x_i x_i^\top.
$$

Hence,

$$
\sigma_{\min}^2 \|x\|_{(\sum_{i=1}^K \lambda_i x_i x_i^\top)^{-1}}^2 \leq \|x\|_{A(\lambda)^{-1}}^2 \leq \sigma_{\max}^2 \|x\|_{(\sum_{i=1}^K \lambda_i x_i x_i^\top)^{-1}}^2.
$$

Applying this to the result of Theorem 4.1 and invoking Lemma E.8, we see that

$$
\sigma_{\min}^2 \min_{\lambda \in P_{\mathcal{X}}} \max_{h \in \mathcal{H}_{\mathrm{OBJ}}, q \in \mathcal{Q}_{\mathrm{OBJ}}} \frac{\|h-q\|_{(\sum_{x \in \mathcal{X}} \lambda_x x x^\top)^{-1}}^2}{\{(h-q)^\top \theta^* - b_{\mathrm{OBJ}}\}^2}
$$

$$
\leq \min_{\lambda \in P_{\mathcal{X}}} \max_{h \in \mathcal{H}_{\mathrm{OBJ}}, q \in \mathcal{Q}_{\mathrm{OBJ}}} \frac{\|h-q\|_{A(\lambda)^{-1}}^2}{\{(h-q)^\top \theta^* - b_{\mathrm{OBJ}}\}^2}
$$

$$
\leq \sigma_{\max}^2 \min_{\lambda \in P_{\mathcal{X}}} \max_{h \in \mathcal{H}_{\mathrm{OBJ}}, q \in \mathcal{Q}_{\mathrm{OBJ}}} \frac{\|h-q\|_{(\sum_{x \in \mathcal{X}} \lambda_x x x^\top)^{-1}}^2}{\{(h-q)^\top \theta^* - b_{\mathrm{OBJ}}\}^2}.
$$

Recalling that

$$
\rho_{\mathrm{OBJ}}^* := \sigma_{\max}^2 \min_{\lambda \in P_{\mathcal{X}}} \max_{h \in \mathcal{H}_{\mathrm{OBJ}}, q \in \mathcal{Q}_{\mathrm{OBJ}}} \frac{\|h-q\|_{(\sum_{x \in \mathcal{X}} \lambda_x x x^\top)^{-1}}^2}{\{(h-q)^\top \theta^* - b_{\mathrm{OBJ}}\}^2}
$$

is the lower bound in both the best arm and level set problems for the homoskedastic variance algorithm in a heteroskedastic noise setting (when upper bounding the variance), we see that

$$\kappa^{-1} \le \psi^*_{\text{OBJ}}/\rho^*_{\text{OBJ}} \le 1.$$

Moreover, both the upper and lower bounds are tight by taking $\sigma_x^2 = \sigma_{\min}^2$, $\forall x \in \mathcal{X}$ (to match the lower bound) and $\sigma_x^2 = \sigma_{\max}^2$, $\forall x \in \mathcal{X}$ (to match the upper bound). ∎

# H    Experiment Details

This Appendix gives more details on the experiments presented in Section 5.

## H.1    Oracle Designs

We demonstrate the difference in oracle allocations referenced by Experiments 1-3 in Section 5. We begin with Experiment 1 in Fig. 4, where the oracle distribution that assumes homoskedastic variance devotes more attention to the informative directions with smaller magnitude. In contrast, the heteroskedastic oracle balances evenly because each informative arm has an equivalent signal-noise ratio when accounting for variance.

Next, we analyze Experiment 2 in Fig. 5. In this setting the oracle distribution that assumes homoskedastic variance allocates evenly between three informative arms, while the heteroskedastic oracle prioritizes the informative directions with less variance.

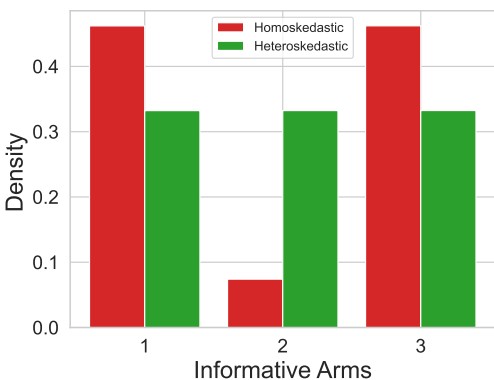
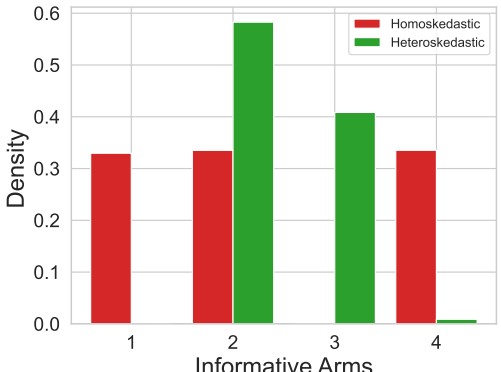

Figure 4: In Experiment 1, the Heteroskedastic Oracle balances evenly between the informative arms.

Figure 5: In Experiment 2, the Heteroskedastic Oracle prioritizes informative arms with less variance.

Lastly, we examine the oracle distributions for the multivariate testing experiment in Fig. 6. Note that the first four arm have higher variance than the last four arms. We can see in Fig. 6 that the Heteroskedastic Oracle prioritizes the second group of arms because these will require more samples to identify their effect sizes.

## H.2    Assuming Homosekedastic Noise

If we assume that there is homoskedastic noise in a heteroskedastic setting, then we need to upper-bound the noise with $\sigma_{\max}^2$ in the sample size calculation of Alg. 2 in order to maintain correctness. In Section 5, we compare H-RAGE to RAGE; the latter uses the $\sigma_{\max}^2$ upper-bound and is described in Alg. 5.

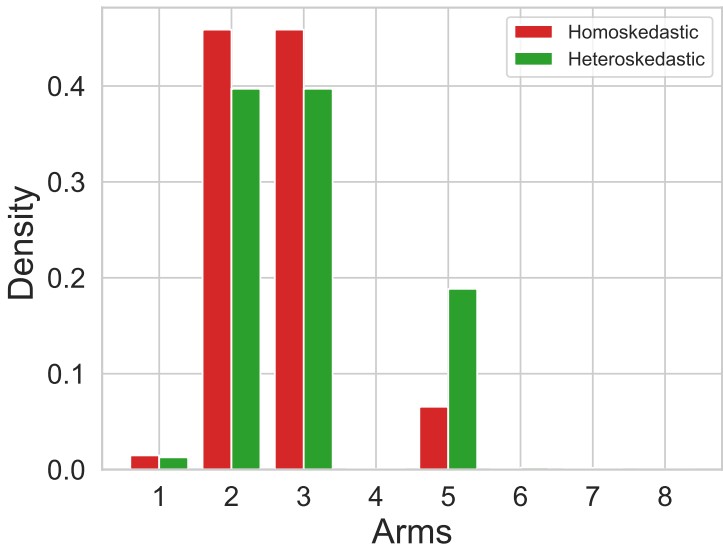

Figure 6: In the multivariate testing experiment, the Heteroskedastic Oracle prioritizes the arms that have higher variance, i.e., the arms that include the second variation in the first dimension.

---

**Algorithm 5:** (RAGE) Randomized Adaptive Gap Elimination

---

**Result:** Find $z^* := \arg\max_{z \in \mathcal{Z}} z^\top \theta^*$ for BAI or $G_\alpha := \{z \in \mathcal{Z} : z^\top \theta^* > \alpha\}$ for LS

1 **Input:** $\mathcal{X} \in \mathbb{R}^d$, $\mathcal{Z} \in \mathbb{R}^d$, confidence $\delta \in (0,1)$, OBJ $\in$ {BAI,LS}, threshold $\alpha \in \mathbb{R}$

2 **Initialize:** $\ell \leftarrow 1$, $\mathcal{Z}_1 \leftarrow \mathcal{Z}$, $\mathcal{G}_1 \leftarrow \emptyset$, $\mathcal{B}_1 \leftarrow \emptyset$

3 **while** *($|\mathcal{Z}_\ell| > 1$ and OBJ=BAI) or ($|\mathcal{Z}_\ell| > 0$ and OBJ=LS)* **do**

4      Let $\widehat{\lambda}_\ell \in P_\mathcal{X}$ be a minimizer of $q\{\lambda, \mathcal{Y}(\mathcal{Z}_\ell)\}$ if OBJ=BAI and $q(\lambda, \mathcal{Z}_\ell)$ if OBJ=LS where

$$q(\mathcal{V}) = \inf_{\lambda \in P_\mathcal{X}} q(\lambda; \mathcal{V}) = \inf_{\lambda \in P_\mathcal{X}} \max_{z \in \mathcal{V}} ||z||^2_{(\sum_{x \in \mathcal{X}} \lambda_x x x^\top)^{-1}}$$

5      Set $\epsilon_\ell = 2^{-\ell}$, $\tau_\ell = 2\epsilon_\ell^{-2}\sigma^2_{\max}q(\mathcal{Z}_\ell)\log(8\ell^2|\mathcal{Z}|/\delta)$

6      Pull arm $x \in \mathcal{X}$ exactly $\lceil \tau_\ell \widehat{\lambda}_{\ell,x} \rceil$ times for $n_\ell$ samples and collect $\{x_{\ell,i}, y_{\ell,i}\}_{i=1}^{n_\ell}$

7      Define $A_\ell := \sum_{i=1}^{n_\ell} x_{\ell,i} x_{\ell,i}^\top$, $b_\ell = \sum_{i=1}^{n_\ell} x_{\ell,i} y_{\ell,i}$ and construct $\widehat{\theta}_\ell = A_\ell^{-1} b_\ell$

8      **if** *OBJ is BAI* **then**

9          $\mathcal{Z}_{\ell+1} \leftarrow \mathcal{Z}_\ell \backslash \{z \in \mathcal{Z}_\ell : \max_{z' \in \mathcal{Z}_\ell} \langle z' - z, \widehat{\theta}_\ell \rangle > \epsilon_\ell\}$

10      **else**

11          $\mathcal{G}_{\ell+1} \leftarrow \mathcal{G}_\ell \cup \{z \in \mathcal{Z}_\ell : \langle z, \widehat{\theta}_\ell \rangle - \epsilon_\ell > \alpha\}$

12          $\mathcal{B}_{\ell+1} \leftarrow \mathcal{B}_\ell \cup \{z \in \mathcal{Z}_\ell : \langle z, \widehat{\theta}_\ell \rangle + \epsilon_\ell < \alpha\}$

13          $\mathcal{Z}_{\ell+1} \leftarrow \mathcal{Z}_\ell \backslash \{\{\mathcal{G}_\ell \backslash \mathcal{G}_{\ell+1}\} \cup \{\mathcal{B}_\ell \backslash \mathcal{B}_{\ell+1}\}\}$

14      **end**

15      $\ell \leftarrow \ell + 1$

16 **end**

17 **Output:** $\mathcal{Z}_\ell$ for BAI or $\mathcal{G}_\ell$ for LS

