# OpenReview forum: "Experimental Designs for Heteroskedastic Variance"
_NeurIPS.cc/2023/Conference — NeurIPS 2023 poster_

### Official Review · Reviewer_Jphn · 2023-06-23

**Soundness:** 4 excellent
**Presentation:** 4 excellent
**Contribution:** 3 good
**Rating:** 7
**Confidence:** 1

**Summary:**

This paper studies linear experimental design in the presence of heteroskedastic noise. The model can observe noise linear measurements generated by unknown model parameters, with sub-Gaussian covariance depends on the measurement vector. The goal is to estimate the unknown covariance matrix and corresponding covariance wrt measurement $x$. The algorithm works as follows: it uses the first half of the samples to construct a least-squares estimator for the model parameter, then use the second half to estimate the noises and consequently, the covariance. This algorithm leads to a sample complexity depends quadratically on $d$. As applications, the proposed approach can be used for best arm identification and level set identification. Finally, experiments are conducted to empirically validate the algorithm.

**Strengths:**

This paper studied an interesting problem, in practical setting, heteroskedastic noises usually present, so it is important to study the sample complexity and efficient algorithm for this setting.

The presentation of the paper is good, it is a good mixture of motivation, intuition and technicality. I enjoyed reading the paper.

The results obtained in this paper are strong.

**Weaknesses:**

It will be better to also include the runtime complexity of the proposed algorithm. Linear bandits now admit highly efficient algorithms with curated data structures [Yang et al., ICML'22] and so does efficient experimental design [Allen-Zhu et al., ICML'17]. It will be interesting to study the runtime complexity of the proposed algorithm.

[Yang el al., ICML'22]: S. Yang, T. Ren, S. Shakkottai, E. Price, I. Dhillion, S. Sanghavi. Linear Bandit Algorithms with Sublinear Time Complexity.

[Allen-Zhu et al., ICML'17]: Z. Allen-Zhu, Y. Li, A. Singh, Y. Wang. Near-Optimal Discrete Optimization for Experimental Design: A Regret Minimization Approach.

**Questions:**

What is the runtime complexity of the proposed algorithm?

**Limitations:**

Yes.

---

> ### Author Rebuttal · Authors · 2023-08-09
>
> Thank you for your careful review and positive comments. We will cite the papers you listed.
>
> The time complexity depends on the Frank-Wolfe method. Defining $k\in \mathbb{N}$ as the number of steps taken by the procedure, the algorithm has a worst case convergence rate of $O(1/k)$ [Jaggi, 2013]. Consequently, the time complexity to achieve a distribution within $\epsilon$ of the optimal value is $O\left [ \left (|\mathcal{Z}|d^2 + |\mathcal{Z}|^2d + |\mathcal{X}|d^2 \right ) /\epsilon \right ]$. This is not a major concern in practice since the optimal distribution is computed in rounds and can be done offline, while online we can simply sample from the distribution.

---

> > ### Comment · Reviewer_Jphn · 2023-08-10
> >
> > Thank you for your response! You mentioned the runtime complexity mainly depends on Frank-Wolfe, I wonder whether recent developments on data-structure-based Frank-Wolfe methods would lead to a more efficient implementation of your algorithm: [Xu et al., NeurIPS'21] and [Song et al., Arxiv'22]. These papers show that by cleverly using data-structures, the complexity of Frank-Wolfe can be improved beyond an exhaustive search over the entire convex hull.
> >
> > [Xu et al., NeurIPS'21] Zhaozhuo Xu, Zhao Song and Anshumali Shrivastava. Breaking the Linear Iteration Cost Barrier for Some Well-known Conditional Gradient Methods Using MaxIP Data-structures.
> >
> > [Song et al., Arxiv'22] Zhao Song, Zhaozhuo Xu, Yuanyuan Yang and Lichen Zhang. Accelerating Frank-Wolfe Algorithm using Low-Dimensional and Adaptive Data Structures.

---

> > > ### Author Response · Authors · 2023-08-14
> > >
> > > This is an interesting question. The traditional Frank-Wolfe algorithm steps in the direction of the point on the convex hull of the constraint boundary that has the smallest inner product with the gradient. Both the papers you reference focus on the use of efficient data structures to find this direction. In our case of the simplex, this step reduces to finding the smallest entry of the gradient. Thus, it is not clear that additional efficiency gains are possible using the ideas presented in the referenced papers.

---

> > > > ### Comment · Reviewer_Jphn · 2023-08-14
> > > >
> > > > Thank you for your comments, I would suggest to incorporate a more detailed description of your algorithm in the paper. I'll keep my score as is.

---

### Official Review · Reviewer_jG9W · 2023-07-02

**Soundness:** 3 good
**Presentation:** 3 good
**Contribution:** 3 good
**Rating:** 6
**Confidence:** 2

**Summary:**

This paper investigates the problem of linear regression under heteroskedastic noise with adaptive data collection.
The proposed algorithm begins with estimating the covariance structure via Alg 1 and then using this in Alg 2 to construct an optimal design for estimating $\theta$. The algorithm is instance-optimal up to an additive term depending on the conditioning $\kappa$. They conclude with experiments demonstrating their algorithm's performance.

**Strengths:**

* This work analyzes a useful problem and proposes a nice adaptation of RAGE to their setup.

**Weaknesses:**

* I'm not sure how common or realistic this assumption is, but the heteroskedasticity model (i.e., that $\sigma_x^2 = x^T\Sigma^\star x$) seems very restrictive to me. It would be nice to have some further discussion about this and possibly about extending the algorithm to more general noise settings.
* As they remark in the paper, the lower bound is weakened by the fact that the noise variances are considered to be known, and so the lower bound essentially becomes the same as Fiez et al. The lower bound would be much more compelling to me if it could incorporate the difficulty associated with estimating $\Sigma^\star$ (which could also address the current $\kappa$-dependent gap between the upper and lower bounds).

**Questions:**

* Regarding the optimality of Alg. 2, what is the appropriate regime to be thinking of for the quantities in Theorems 1 and 2? For example, how should the $\log(\Delta^{-1})\log(|Z|/\delta)$ multiplicative factor differing between the upper and lower bounds compare against $\psi^\star$. Also, how large will $d$ be, given that the additive suboptimality terms has a $d^2$ factor in it?



Minor:
* In Alg. 1, should it say $\lambda^\star_x$ in line 4 and $\lambda^\dag_x$ in line 8? Otherwise, I don't see how the quantities computed in lines 3 and 7 are used.

**Limitations:**

The authors adequately addressed the limitations.

---

> ### Author Rebuttal · Authors · 2023-08-09
>
> Thank you for your careful review and positive comments.
>
> - In the Related Works Section, we cite several papers in the active learning setting that have used this noise model [Chaudhuri et al., 2017, Soare et al., 2013]. Additionally, our heteroskedastic variance structure arises in linear mixed effects models [Huang et al., 2022], which are used in a variety of experimental settings to account for multi-level heterogeneity (patient, spatial and block design variation [Oberg and Mahoney, 2007]). In Section 1.1, we will note the connection to the linear mixed effects model covariance structure. Lastly, the paper mentions that our heteroskedastic variance model generalizes the standard multi-armed bandit to a setting where each arm has its own variance (in this case, the measurement vectors, $\mathcal{X}$, are the standard bases and $\Sigma^*$ is diagonal).
> - To address your question, the $\log(\Delta^{-1})\log(|Z|/\delta)$ multiplicative factor results from running the algorithm in rounds and is present in the homoskedastic setting as well. It should be small compared to $\psi^*$ as long as the problem is sufficiently hard because $\psi^*$ depends polynomially on the gap that is most difficult to learn (as opposed to logarithmically on the smallest gap and arm set size).  The dimension $d$ is an instance-dependent parameter. As we argue in the paper, it is natural for the suboptimality term to depend on $d^2$ since this is an instance dependent parameter that scales with the degrees of freedom in $\Sigma^*$ and the lower bound assumes that $\Sigma^{\ast}$ is known.

---

> > ### Comment · Reviewer_jG9W · 2023-08-18
> > **Response to authors**
> >
> > Thank you for your response, I will maintain my current score.

---

### Official Review · Reviewer_VoEg · 2023-07-06

**Soundness:** 3 good
**Presentation:** 3 good
**Contribution:** 3 good
**Rating:** 7
**Confidence:** 4

**Summary:**

The paper considers the problem of Experimental Designs for Heteroskedastic Variance. Here, a learner has access to finite set of measurement vectors $\mathcal{X}$ and may query observations of the following form for chosen $x \in \mathcal{X}$: $y = x^\top \theta^* + \eta$ where $\eta$ is mean-zero and sub-Gaussian with variance $\sigma^2_x = x^\top \Sigma x$. The paper first designs a procedure to estimate $\Sigma$ and then utilizes this estimate for two related problems featuring linear objectives: Best-arm Identification and Level-set Identification. The paper then shows that when heteroskedastic variance is present, this procedure leads to improved results for these tasks. The paper then conducts empirical experiments to validate their claims.

The main technical contribution of the paper is the procedure to estimate $\Sigma$. If $\theta$ were known, $(y - \theta^\top x)^2$ provides an unbiased estimate of $\sigma_x^2$. However, since $\theta^*$ is not known, the paper first estimates it via a weighted least-squares estimator where the weights are chosen to ensure that each $x \in \mathcal{X}$ is well-represented in the spectrum of the weighted second-moment matrix. This ensures that $\theta$ is accurate (at least along the $x$ in $\mathcal{X}$). Then, a similar sampling procedure is used with the vectors $\phi_x = \mathrm{vec} (xx^\top)$ and noting that $(y - x^\top \theta^*)^2$ is an unbiased estimate of $\sigma_x^2 = \mathrm{vec} (xx^\top)^\top \mathrm{vec} (\Sigma)$ which is a linear equation in $\Sigma$. A similar weighting procedure is used to obtain an estimate of $\Sigma$ such that $\sigma_x^2$ is well-approximated for any $x \in \mathcal{X}$. These estimates are then used to design improved strategies for the aforementioned estimation tasks with linear objects.

Overall this is a nice paper with strong technical results. The setting seems novel and interesting and the technical analysis is non-trivial. The main question I have for the authors is that the sampling procedure incurs large sample complexities in some extreme cases. Say, $\mathcal{X}$ corresponded to the standard basis vectors. In this setting, the error bound on estimating $\sigma_{e_1}$ depends on $\max_j \sigma_{e_j}$ while intuitively this dependence seems extreme. Furthermore, the fact that $d^2$ samples are needed to estimate $\Sigma$ only along the directions in $\mathcal{X}$ also seems too large if $\mathcal{X}$ is not too large. It would be great if the authors can comment on these dependencies.


**Strengths:**

See main review

**Weaknesses:**

See main review

**Questions:**

See main review

**Limitations:**

Yes

---

> ### Author Rebuttal · Authors · 2023-08-09
>
> Thank you for your careful review and positive comments.
>
> - This is a good point. We focus on a uniform bound over variance estimates because this is necessary for the best-arm and level set identification algorithms. In these procedures, the uniform bound quantifies the error in both the experimental design and the weighted least squares estimator. We then determine the number of samples needed to control this bound to identify the correct arm (or arm set) with $1-\delta$ probability.
> - In paradigms where $|\mathcal{X}| < d^2$, there is not enough information to accurately estimate $\Sigma^*$. Consequently, it may be advantageous to ignore the structure and estimate the variances of the arms separately as Fontaine et al. [2021]  does. We could then use these estimates in the best-arm and level set identification algorithms with their corresponding bounds and maintain the correct identification rate. However, we focus on the case where $|\mathcal{X}| >> d^2$, which aligns with a majority of the literature.

---

> > ### Comment · Reviewer_VoEg · 2023-08-18
> > **Reviewer Response**
> >
> > Thank you for the response. I will retain my current evaluation.

---

### Official Review · Reviewer_mrcD · 2023-07-07

**Soundness:** 2 fair
**Presentation:** 3 good
**Contribution:** 2 fair
**Rating:** 3
**Confidence:** 4

**Summary:**

This paper considers the online linear experimental design problem that is affected by heteroskedastic noise. The authors propose a two-phase sample splitting method aimed at estimating the unknown parameters of noise variance, utilizing the principle of G-optimal experimental designs.

Furthermore, the study extends the application of the proposed methodology to handle fixed confidence transductive best-arm and level set identification problems under conditions of heteroskedastic noise. The paper also deals with the instance-dependent lower bounds.

**Strengths:**

This paper does solid work and is in relatively good shape.

**Weaknesses:**

1. The problem is not well motivated. It is unclear what are the practical applications of this problem.

2. The numerical experiments only contain some toy examples. It is unclear how it performed in a large-scale set.

3. Comparision between the previous works are not present.

4. The problem setting looks weird. The sensing noise variance is $x^T \Sigma x$, why it depends on the sensing vector $x$? What are the motivations for this problem setting?

**Questions:**

1. How is the current approach compared to the previous works? Both in a theoretical sense and numerical experiments.

2. What motivates the problem setting, that is, the sensing noise variance is $x^T \Sigma x$.

**Limitations:**

See above.

---

> ### Author Rebuttal · Authors · 2023-08-09
>
> Thank you for your constructive comments. It is encouraging to hear that the paper does solid work and is in relatively good shape.
>
> - Motivation for the Problem- The problem is motivated by: (i) the presence of heteroskedastic errors in numerous applications; and (ii) the significant impact of heteroskedastic error on the performance of methods which do not adequately account for it. First, we illustrate the ubiquity of heteroskedasticity with examples such as e-commerce [Kaptein and Parvinen, 2015] and the volatility of stocks across sectors [Dai et al., 2022]. Heteroskedasticity is also commonly encountered in clinical data [Grissom, 2000, Zhou et al., 2023], spatial data [Yan et al., 2022, Sato and Matsuda, 2021], and genomics [Ting et al., 2023, Dumitrascu et al., 2019] among other areas. We then show that algorithms in the best-arm and level set identification setting improve dramatically when they account for heteroskedastic variances. This is demonstrated theoretically and empirically in Sections 1, 4 and 5.
> - Numerical Experiments- In our simulations (Section 5), the first two settings are standard benchmark examples that illustrate important features of the algorithm [Soare et al., 2014]. The third setting considers Multivariate Testing, which is a relevant and realistic paradigm in advertising [Hill et al., 2017].
> - Comparison- We place our work in the literature in Section 2 (Related Works). We also make comparisons in terms of theory (Sections 3 and 4) and numerical experiments (Sections 1, 3 and 5). The theoretical comparisons of Sections 1 and 4 involve $\kappa$, which quantifies the level of heteroskedasticity. We show that the state of the art suffers from a multiplicative dependency on $\kappa$, while our method only has an additive dependency. In Sections 1 and 5, we compare H-RAGE to RAGE empirically.
> - Motivation for the Noise Structure- In the Related Works Section, we cite several papers in the active learning setting that have used this noise model [Chaudhuri et al., 2017, Soare et al., 2013]. Additionally, our heteroskedastic variance structure arises in linear mixed effects models [Huang et al., 2022], which are used in a variety of experimental settings to account for multi-level heterogeneity (patient, spatial and block design variation [Oberg and Mahoney, 2007]). In Section 1.1, we will note the connection to the linear mixed effects model covariance structure. Lastly, the paper mentions that our heteroskedastic variance model generalizes the standard multi-armed bandit to a setting where each arm has its own variance (in this case, the measurement vectors, $\mathcal{X}$, are the standard bases and $\Sigma^*$ is diagonal).

---

> > ### Comment · Reviewer_mrcD · 2023-08-20
> > **Thank you**
> >
> > Dear author,
> >
> > Thanks for your response. I would like to remain my previous rating. The reason is as follows.
> >
> > 1. The numerical experiments are very limited. Only FIgure 3 is presented to test the theory. Also, these examples are quite small, basically some toy examples. There are no large-scale experiments to evaluate and test your theory.
> >
> > 2. The theoretical contribution is very limited. As a researcher specializing in theory, I am quite confident to say that the proof techniques in this paper are quite standard. I am afraid that there is not enough contribution in this direction as well.
> >
> > In summary, this paper not only lacks the theoretical contributions but also enough numerical experiments to prove the effectiveness of its algorithm in reality.

---

### Decision · Program_Chairs · 2023-09-21

**Decision:**

Accept (poster)

**Comment:**

Most reviewers felt this was a technically solid paper. It considered experimental design for regression problems for the noise for covariate x is of the form x^TSigmax for some unknown Sigma. It uses half of its allocated samples to learn Sigma and the rest to learn the parameter vector. The paper draws on techniques from the linear bandits literature (specifically for the “pure exploration problem”). Most reviewers felt the paper was well-written and provided interesting theoretical results, although it could have done a better job motivating the problem.